# Optimizing Backward Policies in GFlowNets via Trajectory Likelihood Maximization

**Timofei Gritsaev**
HSE University
Constructor University, Bremen
tgritsaev@gmail.com

**Nikita Morozov**
HSE University
nvmorozov@hse.ru

**Sergey Samsonov**
HSE University
svsamsonov@hse.ru

**Daniil Tiapkin**
CMAP – CNRS – École polytechnique – Institut Polytechnique de Paris
Université Paris-Saclay, CNRS, Laboratoire de mathématiques d'Orsay
daniil.tiapkin@polytechnique.edu

## Abstract

Generative Flow Networks (GFlowNets) are a family of generative models that learn to sample objects with probabilities proportional to a given reward function. The key concept behind GFlowNets is the use of two stochastic policies: a forward policy, which incrementally constructs compositional objects, and a backward policy, which sequentially deconstructs them. Recent results show a close relationship between GFlowNet training and entropy-regularized reinforcement learning (RL) problems with a particular reward design. However, this connection applies only in the setting of a fixed backward policy, which might be a significant limitation. As a remedy to this problem, we introduce a simple backward policy optimization algorithm that involves direct maximization of the value function in an entropy-regularized Markov Decision Process (MDP) over intermediate rewards. We provide an extensive experimental evaluation of the proposed approach across various benchmarks in combination with both RL and GFlowNet algorithms and demonstrate its faster convergence and mode discovery in complex environments.

## 1 Introduction

Generative Flow Networks (GFlowNets, Bengio et al., 2021) are models designed to sample compositional discrete objects, e.g., graphs, from distributions defined by unnormalized probability mass functions. They operate by constructing an object through a sequence of stochastic transitions defined by a *forward policy*. This policy is trained to match the marginal distribution over constructed objects with the target distribution of interest. Since this marginal distribution is generally intractable, an auxiliary *backward policy* is introduced, and the problem is reduced to one of matching distributions over complete trajectories, bearing similarities with variational inference (Malkin et al., 2023).

GFlowNets have found success in various areas, such as biological sequence design (Jain et al., 2022), material discovery (Hernandez-Garcia et al., 2023), molecular optimization (Zhu et al., 2024), recommender systems (Liu et al., 2024), large language model (LLM) and diffusion model fine-tuning (Hu et al., 2023; Venkatraman et al., 2024; Uehara et al., 2024; Zhang et al., 2025), neural architecture search (Chen & Mauch, 2023), combinatorial optimization (Zhang et al., 2023), and causal discovery (Atanackovic et al., 2024).

Theoretical foundations of GFlowNets have been established in the seminal works of Bengio et al. (2021; 2023). Most of the literature has since focused on practical applications of these models, leaving their theoretical properties largely unexplored, with a few exceptions (Krichel et al., 2024; Silva et al., 2024). However, a recent line of works has brought attention to connections between GFlowNets and reinforcement learning (Tiapkin et al., 2024; Mohammadpour et al., 2024; Deleu et al., 2024; He et al., 2024a), showing that the GFlowNet learning problem is equivalent to a specific RL problem with entropy regularization (also called soft RL, Neu et al., 2017; Geist et al., 2019). This opened a new perspective for understanding GFlowNets. The importance of these findings is supported by empirical evidence, as various RL algorithms have proven useful for improving GFlowNets (Tiapkin et al., 2024; Morozov et al., 2024; Lau et al., 2025).

However, these connections still have a limitation related to GFlowNet backward policies. While GFlowNets can be trained with a fixed backward policy, standard GFlowNet algorithms allow the training of the backward policy together with the forward policy (Bengio et al., 2023; Malkin et al., 2022; Madan et al., 2023), resulting in faster convergence of the optimization process. Other algorithms for optimizing backward policies have been proposed in the literature as well (Mohammadpour et al., 2024; Jang et al., 2024), showing benefits for GFlowNet performance. The theory connecting GFlowNets and entropy-regularized RL is based on using the backward policy to add a "correction" to GFlowNet rewards and shows the equivalence between two problems only when the backward policy is fixed. Thus, understanding the backward policy optimization remains a missing piece of this puzzle. Moreover, Tiapkin et al. (2024) demonstrated that this theoretical gap has practical relevance, as optimizing the backward policy using the same RL objective as the forward policy can either fail to improve or even slow down convergence, highlighting the need for a more refined approach.

In this study, we introduce *the trajectory likelihood maximization* (`TLM`) approach for backward policy optimization, which can be integrated with any existing GFlowNet method, including entropy-regularized RL approaches.

We first formulate the GFlowNet training problem as a unified objective involving both forward and backward policies. We then propose an alternating minimization procedure consisting of two steps: (1) maximizing the backward policy likelihood of trajectories sampled from the forward policy and (2) optimizing the forward policy within an entropy-regularized Markov decision process that corresponds to the updated backward policy. The latter step can be achieved by any existing GFlowNet or soft RL algorithm, as it was outlined by Deleu et al. (2024). By approximating these two steps through a single stochastic gradient update, we derive an adaptive approach for combining backward policy optimization with any GFlowNet method, *including soft RL methods.*

Our main contributions are as follows:

- We derive the trajectory likelihood maximization (`TLM`) method for backward policy optimization.
- The proposed method represents the first unified approach for adaptive backward policy optimization in soft RL-based GFlowNet methods. The method is easy to implement and can be integrated with any existing GFlowNet training algorithm.
- We provide an extensive experimental evaluation of `TLM` in four tasks, confirming the findings of Mohammadpour et al. (2024), which emphasize the benefits of training the backward policy in a complex environment with less structure.

Source code: github.com/tgritsaev/gflownet-tlm.

## 2 BACKGROUND

### 2.1 GFLOWNETS

We aim at sampling from a probability distribution over a finite discrete space $\mathcal{X}$ that is given as an unnormalized probability mass function $\mathcal{R} \colon \mathcal{X} \to \mathbb{R}_{\geq 0}$, which we call the *GFlowNet reward*. We denote $Z = \sum_{x \in \mathcal{X}} \mathcal{R}(x)$ to be an (unknown) normalizing constant.

To formally define a generation process in GFlowNets, we introduce a directed acyclic graph (DAG) $\mathcal{G} = (\mathcal{S}, \mathcal{E})$, where $\mathcal{S}$ is a state space and $\mathcal{E} \subseteq \mathcal{S} \times \mathcal{S}$ is a set of edges (or transitions). There is exactly one state, $s_0$, with no incoming edges, which we refer to as the *initial state*. All other states can be reached from $s_0$, and the set of *terminal states* with no outgoing edges coincides with the space of interest $\mathcal{X}$. Non-terminal states $s \notin \mathcal{X}$ correspond to "incomplete" objects and edges $s \to s'$ represent adding "new components" to such objects, transforming $s$ into $s'$. Let $\mathcal{T}$ denote the set of all complete trajectories $\tau = (s_0, s_1, \ldots, s_{n_\tau})$ in the graph, where $\tau$ is a sequence of states such that $(s_i \to s_{i+1}) \in \mathcal{E}$ and that starts at $s_0$ and finishes at some terminal state $s_{n_\tau} \in \mathcal{X}$. As a result, any complete trajectory can be viewed as a sequence of actions that constructs the object corresponding to $s_{n_\tau}$ starting from the "empty object" $s_0$.

We say that a state $s'$ is a child of a state $s$ if there is an edge $(s \to s') \in \mathcal{E}$. In this case, we also say that $s$ is a parent of $s'$. Next, for any state $s$, we introduce the *forward policy*, denoted by $\mathcal{P}_{\mathrm{F}}(s'|s)$ for $(s \to s') \in \mathcal{E}$, as an arbitrary probability distribution over the set of children of the state $s$. In a similar fashion, we define the *backward policy* as an arbitrary probability distribution over the parents of a state $s$ and denote it as $\mathcal{P}_{\mathrm{B}}(s'|s)$, where $(s' \to s) \in \mathcal{E}$.

Given these two definitions, the main goal of GFlowNet training is a search for a pair of policies such that the induced distributions over complete trajectories in the forward and backward directions coincide:

$$\prod_{t=1}^{n_\tau} \mathcal{P}_{\mathrm{F}}(s_t \mid s_{t-1}) = \frac{\mathcal{R}(s_{n_\tau})}{\mathrm{Z}} \prod_{t=1}^{n_\tau} \mathcal{P}_{\mathrm{B}}(s_{t-1} \mid s_t), \quad \forall \tau \in \mathcal{T}. \tag{1}$$

The relation (1) is known as the *trajectory balance constraint* (Malkin et al., 2022). We refer to the left and right-hand sides of (1) as to the forward and backward trajectory distributions and denote them as

$$\mathsf{P}_{\mathcal{T}}^{\mathcal{P}_{\mathrm{F}}}(\tau) := \prod_{i=1}^{n_\tau} \mathcal{P}_{\mathrm{F}}(s_i | s_{i-1}), \qquad \mathsf{P}_{\mathcal{T}}^{\mathcal{P}_{\mathrm{B}}}(\tau) := \frac{\mathcal{R}(s_{n_\tau})}{\mathrm{Z}} \cdot \prod_{i=1}^{n_\tau} \mathcal{P}_{\mathrm{B}}(s_{i-1} | s_i), \tag{2}$$

where $\tau = (s_0, s_1, \ldots, s_{n_\tau}) \in \mathcal{T}$. If the condition (1) is satisfied for all complete trajectories, sampling a trajectory in the forward direction using $\mathcal{P}_{\mathrm{F}}$ will result in a terminal state being sampled with probability $\mathcal{R}(x)/\mathrm{Z}$. We will call such $\mathcal{P}_{\mathrm{F}}$ a *proper* GFlowNet forward policy.

In practice, we train a model (usually a neural network) that parameterizes the forward policy (and possibly other auxiliary functions) to minimize an objective function that enforces the constraint (1) or its equivalent. The main existing objectives are *Trajectory Balance* (`TB`, Malkin et al., 2022), *Detailed Balance* (`DB`, Bengio et al., 2023) and *Subtrajectory Balance* (`SubTB`, Madan et al., 2023). The `SubTB` objective is defined as

$$\mathcal{L}_{\mathrm{SubTB}}(\theta; \tau) = \sum_{0 \le j < k \le n_\tau} w_{jk} \left( \log \frac{F_\theta(s_j) \prod_{t=j+1}^{k} \mathcal{P}_{\mathrm{F}}(s_t | s_{t-1}, \theta)}{F_\theta(s_k) \prod_{t=j+1}^{k} \mathcal{P}_{\mathrm{B}}(s_{t-1} | s_t, \theta)} \right)^2, \tag{3}$$

where $F_\theta(s)$ is a neural network that approximates the *flow* function of the state $s$, see (Bengio et al., 2023; Madan et al., 2023) for more details on the flow-based formalization of the GFlowNet problem. Here $F_\theta(s)$ is substituted with $\mathcal{R}(s)$ for terminal states $s$, and $w_{jk}$ is usually taken to be $\lambda^{k-j}$ and then normalized to sum to 1. `TB` and `DB` objectives can be viewed as special cases of (3), which are obtained by only taking the term corresponding to the full trajectory or to individual transitions, respectively. All objectives allow either training the model in an on-policy regime using the trajectories sampled from $\mathcal{P}_{\mathrm{F}}$ or in an off-policy mode using the replay buffer or some exploration techniques. In addition, it is possible to either optimize $\mathcal{P}_{\mathrm{B}}$ along with $\mathcal{P}_{\mathrm{F}}$ or to use a fixed $\mathcal{P}_{\mathrm{B}}$, e.g., the uniform distribution over parents of each state. One can show that given any fixed $\mathcal{P}_{\mathrm{B}}$, there exists a unique $\mathcal{P}_{\mathrm{F}}$ that satisfies (1); see, e.g., (Malkin et al., 2022).

## 2.2 GFLOWNETS AS SOFT RL

In reinforcement learning (Sutton & Barto, 2018), a typical performance measure of an agent is a *value function*, which is defined as an expected discounted sum of rewards when acting via a given policy. Entropy-regularized reinforcement learning (RL), also known as soft RL (Neu et al. 2017; Geist et al. 2019; Haarnoja et al. 2017), incorporates Shannon entropy $\mathcal{H}(\mathcal{P}_{\mathrm{F}}(\cdot \mid s)) \triangleq \mathbb{E}_{s' \sim \mathcal{P}_{\mathrm{F}}(\cdot | s)}[-\log \mathcal{P}_{\mathrm{F}}(s' \mid s)]$ into the value function. This addition encourages the optimal policy to be more exploratory:

$$V_\lambda^{\mathcal{P}_{\mathrm{F}}}(s; r) \triangleq \mathbb{E}_{\mathcal{P}_{\mathrm{F}}} \left[ \sum_{t=0}^{\infty} \gamma^t \big( r(s_t, s_{t+1}) + \lambda \mathcal{H}(\mathcal{P}_{\mathrm{F}}(\cdot \mid s_t)) \big) \, \Big| \, s_0 = s \right], \tag{4}$$

where $\lambda \ge 0$ is the regularization coefficient. Note that we use the next state instead of the more conventional action representation, as there is a one-to-one correspondence between the action taken and the resulting next state in DAG environments. Similarly, the regularized Q-values $Q_\lambda^{\mathcal{P}_{\mathrm{F}}}(s, s')$ are defined as the expected discounted sum of rewards, augmented by Shannon entropy, given an initial state $s_0 = s$ and the next state $s_1 = s'$. The regularized optimal policy $\mathcal{P}_{\mathrm{F},\lambda}^\star$ is the policy that maximizes $V_\lambda^{\mathcal{P}_{\mathrm{F}}}(s)$ for any state $s$. **Note:** in standard RL notation, a policy is typically denoted as $\pi$. Here, we use $\mathcal{P}_{\mathrm{F}}$, consistently with GFlowNet notation, to avoid notational clutter.

It was proven by Tiapkin et al. (2024) that training a GFlowNet policy $\mathcal{P}_{\mathrm{F}}$ with a fixed $\mathcal{P}_{\mathrm{B}}$ can be formulated as a soft RL problem in a deterministic MDP represented by a directed acyclic graph $\mathcal{G}$, where actions correspond to transitions over edges. For transitions $(s \to x)$ that lead to terminal states, the RL rewards are defined as $r^{\mathcal{P}_{\mathrm{B}}}(s, x) = \log \mathcal{P}_{\mathrm{B}}(s \mid x) + \log \mathcal{R}(x)$, while for intermediate transitions $(s \to s')$, the rewards are $r^{\mathcal{P}_{\mathrm{B}}}(s, s') = \log \mathcal{P}_{\mathrm{B}}(s \mid s')$. By setting $\lambda = 1$ and $\gamma = 1$, it can

be shown that the optimal policy $\mathcal{P}^{\star}_{\mathrm{F},\lambda=1}(\cdot \mid s)$ in this regularized MDP coincides with the proper GFlowNet forward policy $\mathcal{P}_{\mathrm{F}}(\cdot \mid s)$, which is uniquely determined by $\mathcal{P}_{\mathrm{B}}$ and $\mathcal{R}$ (Theorem 1, Tiapkin et al., 2024), thereby inducing the correct marginal distribution over terminal states $\mathcal{R}(x)/\mathrm{Z}$.

In addition, Proposition 1 of Tiapkin et al. (2024) provides a connection between the corresponding regularized value function at the initial state $s_0$ for any forward policy $\mathcal{P}_{\mathrm{F}}$ (not necessarily proper) and KL-divergence between the induced trajectory distributions:

$$V^{\mathcal{P}_{\mathrm{F}}}_{\lambda=1}(s_0; r^{\mathcal{P}_{\mathrm{B}}}) = \log \mathrm{Z} - \mathrm{KL}(\mathsf{P}^{\mathcal{P}_{\mathrm{F}}}_{\mathcal{T}} \| \mathsf{P}^{\mathcal{P}_{\mathrm{B}}}_{\mathcal{T}}).$$

The main practical corollary of these results is that any RL algorithm that works with entropy regularization can be utilized to train GFlowNets when $\mathcal{P}_{\mathrm{B}}$ is fixed. For example, Tiapkin et al. (2024) demonstrated the efficiency of the classical `SoftDQN` algorithm (Haarnoja et al., 2017) and its modified variant called `MunchausenDQN` (Vieillard et al., 2020). Moreover, it turns out that, under this framework, the existing GFlowNet training algorithms can be derived from existing RL algorithms. Tiapkin et al. (2024) showed that on-policy `TB` corresponds to policy gradient algorithms, and `DB` corresponds to a dueling variant of `SoftDQN`. At the same time, Deleu et al. (2024) showed that `TB`, `DB` and `SubTB` algorithms can be derived from path consistency learning (`PCL`, Nachum et al., 2017) under the assumption of a fixed backward policy.

## 2.3 Backward Policies in GFlowNets

The idea of backward policy optimization is essential to understanding the GFlowNets training procedure. In particular, the most straightforward approach used in GFlowNet literature (Malkin et al., 2022; Bengio et al., 2023) proposes to optimize the forward and backward policies directly through the same GFlowNet objective (e.g., (3)). This approach can accelerate the speed of convergence Malkin et al. (2022), at the same time potentially leading to less stable training (Zhang et al., 2022). This phenomenon motivates studying the backward policy optimization in the recent works of Mohammadpour et al. (2024) and Jang et al. (2024).

Mohammadpour et al. (2024) suggested using the backward policy with maximum possible trajectory entropy, thus focusing on the *exploration* challenges of GFlowNets. Such policy is proven to be $\mathcal{P}_{\mathrm{B}}(s \mid s') = n(s)/n(s')$, where $n(s)$ is the number of trajectories which starts at $s_0$ and end at $s$. It corresponds to the uniform one if the number of paths to all parent nodes is equal. When $n(s)$ cannot be computed analytically, Mohammadpour et al. (2024) propose to learn $\log n(s)$, $s \in \mathcal{S}$ alongside the forward policy using its relation to the value function of the soft Bellman equation in the *inverted MDP* (see Definition 2 in Mohammadpour et al. (2024)). Mohammadpour et al. (2024) utilize RL as a tool to find the maximum entropy backward policy and make the connection to RL solely for such policies. In contrast, our work theoretically considers the simultaneous optimization of the forward and the backward policy from the RL perspective, and develops an optimization algorithm grounded in it. The approach of Mohammadpour et al. (2024) showed consistently better results in less structured tasks, like QM9 molecule generation (see Section 4.3 for a detailed description). At the same time, more structured environments with a less challenging exploration counterpart show less benefits of the proposed backward training approach.

In another line of work, Jang et al. (2024) claim that the existing GFlowNets training procedures tend to under-exploit the high-reward objects and propose a Pessimistic Backward Policy approach. Thus, the primary aim of Jang et al. (2024) is to focus on the *exploitation* of the current information about high-reward trajectories. Towards this aim, they focus on maximizing the observed backward flow $\mathsf{P}^{\mathcal{P}_{\mathrm{B}}}_{\mathcal{T}}(\tau)$ (see (2)) for trajectories leading to high-reward objects, which are stored in a replay buffer. Our approach shares similarities with this method as both compute the loss over whole trajectories but differs in theoretical motivation and the choice of trajectories for $\mathcal{P}_{\mathrm{B}}$ optimization.

As a limitation of both Mohammadpour et al. (2024) and Jang et al. (2024), we mention the fact that only a single GFlowNet training objective is used in both papers (`SubTB` and `TB` respectively) to evaluate approaches for backward policy optimization, while we carry out experimental evaluation with various GFlowNet training objectives (see Section 4).

## 3 Trajectory Likelihood Maximization

The objective of our method is to formalize the optimization process for the backward policy for reinforcement learning-based approaches. It is worth mentioning that soft RL methods cannot address

the changing of the reward function, except for reward shaping schemes (Ng et al., 1999) that preserve the total reward of any trajectory. Therefore, we need to return to the underlying GFlowNet problem. Let us look at the following optimization problem:

$$\min_{\mathcal{P}_{\mathrm{F}}\in\Pi_{\mathrm{F}},\mathcal{P}_{\mathrm{B}}\in\Pi_{\mathrm{B}}} \mathrm{KL}(\mathsf{P}_{\mathcal{T}}^{\mathcal{P}_{\mathrm{F}}}\|\mathsf{P}_{\mathcal{T}}^{\mathcal{P}_{\mathrm{B}}}), \tag{5}$$

where $\Pi_{\mathrm{F}}$ and $\Pi_{\mathrm{B}}$ represent the spaces of forward and backward policies, respectively, and $\mathsf{P}_{\mathcal{T}}^{\mathcal{P}_{\mathrm{F}}}$ and $\mathsf{P}_{\mathcal{T}}^{\mathcal{P}_{\mathrm{B}}}$ are defined in (2). It is easy to see that any solution to the problem (5) satisfies the trajectory balance constraint (1) and thus induces a proper GFlowNet forward policy. Additionally, it is worth mentioning that for any fixed $\mathcal{P}_{\mathrm{B}}$ the problem (5) is equivalent to maximizing value

$$V_{\lambda=1}^{\mathcal{P}_{\mathrm{F}}}(s_0; r^{\mathcal{P}_{\mathrm{B}}}) = \log \mathrm{Z} - \mathrm{KL}(\mathsf{P}_{\mathcal{T}}^{\mathcal{P}_{\mathrm{F}}}\|\mathsf{P}_{\mathcal{T}}^{\mathcal{P}_{\mathrm{B}}})$$

over $\mathcal{P}_{\mathrm{F}} \in \Pi_{\mathrm{F}}$ in an entropy-regularized MDP, as established in Tiapkin et al. (2024, Proposition 1). Thus, the joint optimization resembles the RL formulation with non-stationary rewards. To leverage the problem's block structure, we propose a meta-algorithm consisting of two iterative steps, repeated until convergence:

$$\mathcal{P}_{\mathrm{B}}^{t+1} \approx \arg\min_{\mathcal{P}_{\mathrm{B}}} \mathrm{KL}\left(\mathsf{P}_{\mathcal{T}}^{\mathcal{P}_{\mathrm{F}}^{t}}\|\mathsf{P}_{\mathcal{T}}^{\mathcal{P}_{\mathrm{B}}}\right), \qquad \mathcal{P}_{\mathrm{F}}^{t+1} \approx \arg\min_{\mathcal{P}_{\mathrm{F}}} \mathrm{KL}\left(\mathsf{P}_{\mathcal{T}}^{\mathcal{P}_{\mathrm{F}}}\|\mathsf{P}_{\mathcal{T}}^{\mathcal{P}_{\mathrm{B}}^{t+1}}\right). \tag{6}$$

It is worth noting that if these optimization problems are solved exactly, the algorithm converges after the first iteration. This occurs because, for every fixed backward policy $\mathcal{P}_{\mathrm{B}}$, there is a unique forward policy $\mathcal{P}_{\mathrm{F}}$, such that $\mathsf{P}_{\mathcal{T}}^{\mathcal{P}_{\mathrm{F}}} = \mathsf{P}_{\mathcal{T}}^{\mathcal{P}_{\mathrm{B}}}$, see, e.g., (Malkin et al., 2022), ensuring that the loss function reaches its global minimum. In the following sections, we provide implementation details on approximating these two steps.

**First Step: Trajectory Likelihood Maximization.** Using the connection between forward KL divergence minimization and maximum likelihood estimation (MLE), we formulate the following *trajectory likelihood maximization* objective:

$$\theta_{\mathrm{B}}^{t+1} \approx \arg\min_{\theta} \mathbb{E}_{\tau \sim \mathcal{P}_{\mathrm{F}}^{t}}[\mathcal{L}_{\mathtt{TLM}}(\theta; \tau)], \qquad \mathcal{L}_{\mathtt{TLM}}(\theta; \tau) := -\sum_{i=1}^{n_\tau} \log \mathcal{P}_{\mathrm{B}}(s_{i-1}|s_i, \theta). \tag{7}$$

In this formulation, $\tau = (s_0, s_1, \ldots, s_{n_\tau})$ denotes a trajectory generated by the forward policy $\mathcal{P}_{\mathrm{F}}^{t}$. This step seeks to update the backward policy by minimizing the negative log-likelihood of trajectories generated from the forward policy. Additionally, instead of solving (7) for exact $\arg\min$ for every $t$, we perform one stochastic gradient update

$$\theta_{\mathrm{B}}^{t+1} = \theta_{\mathrm{B}}^{t} - \gamma \nabla_{\theta} \mathcal{L}_{\mathtt{TLM}}(\theta_{\mathrm{B}}^{t}; \tau).$$

**Second Step: Non-Stationary Soft RL Problem.** To approximate the second step of (6), we exploit the equivalence between the GFlowNet framework and the entropy-regularized RL problem. This leads to the following expression:

$$\mathcal{P}_{\mathrm{F}}^{t+1} \approx \arg\min_{\mathcal{P}_{\mathrm{F}}\in\Pi_{\mathrm{F}}} \mathrm{KL}\left(\mathsf{P}_{\mathcal{T}}^{\mathcal{P}_{\mathrm{F}}}\|\mathsf{P}_{\mathcal{T}}^{\mathcal{P}_{\mathrm{B}}^{t+1}}\right) \iff \mathcal{P}_{\mathrm{F}}^{t+1} \approx \arg\max_{\mathcal{P}_{\mathrm{F}}\in\Pi_{\mathrm{F}}} V_{\lambda=1}^{\mathcal{P}_{\mathrm{F}}}\left(s_0; r^{\mathcal{P}_{\mathrm{B}}^{t+1}}\right), \tag{8}$$

where $r^{\mathcal{P}_{\mathrm{B}}}$ is the RL reward function corresponding to the backward policy $\mathcal{P}_{\mathrm{B}}$. This step can be solved using any soft RL method, such as `SoftDQN` (Haarnoja et al., 2018). Additionally, it is noteworthy that all existing GFlowNet algorithms with a fixed backward policy can be viewed as variations of existing RL methods, see, e.g., (Deleu et al., 2024). Thus, they can be used to solve the optimization problem in (8).

To mitigate the computational overhead of searching for exact $\arg\min$ in (8), we also propose to perform a single stochastic gradient update in the corresponding GFlowNet training algorithm

$$\theta_{\mathrm{F}}^{t+1} = \theta_{\mathrm{F}}^{t} - \eta \nabla_{\theta} \mathcal{L}_{\mathtt{Alg}}(\theta_{\mathrm{F}}^{t}; \tau, \mathcal{P}_{\mathrm{B}}^{t+1}),$$

where $\mathcal{L}_{\mathtt{Alg}}$ represents the loss function associated with a GFlowNet or soft RL method, such as `SubTB` or `SoftDQN`. Here, $\tau$ denotes a (possibly off-policy) trajectory.

The complete procedure can be interpreted as a soft RL method with changing rewards. Our suggested method is summarized in Algorithm 1 and can be paired with any GFlowNet training method `Alg` (e.g., `DB`, `TB`, `SubTB`, or `SoftDQN`). While for $\mathcal{P}_{\mathrm{B}}$ training our approach uses on-policy trajectories, $\mathcal{P}_{\mathrm{F}}$ can still be trained off-policy, e.g., by utilizing a replay buffer that stores trajectories or transitions.

---

**Algorithm 1** Trajectory Likelihood Maximization

---

1: **Input:** Forward and backward parameters $\theta_{\mathrm{F}}^1, \theta_{\mathrm{B}}^1$, any GFlowNet loss function $\mathcal{L}_{\mathtt{Alg}}$, *(optional)* experience replay buffer $\mathcal{B}$;
2: **for** $t = 1$ **to** $N_{\text{iters}}$ **do**
3:      Sample a batch of trajectories $\{\tau_k^{(t)}\}_{k=1}^K$ from the forward policy $\mathcal{P}_{\mathrm{F}}(\cdot|\cdot, \theta_{\mathrm{F}}^t)$;
4:      *(optional)* Update $\mathcal{B}$ with $\{\tau_k^{(t)}\}_{k=1}^K$;
5:      Update $\theta_{\mathrm{B}}^{t+1} = \theta_{\mathrm{B}}^t - \gamma_t \cdot \frac{1}{K} \sum_{k=1}^K \nabla \mathcal{L}_{\mathtt{TLM}}(\theta_{\mathrm{B}}^t; \tau_k^{(t)})$, see (7);
6:      *(optional)* Resample a batch of trajectories $\{\tau_k^{(t)}\}_{k=1}^K$ from $\mathcal{B}$;
7:      Update $\theta_{\mathrm{F}}^{t+1} = \theta_{\mathrm{F}}^t - \eta_t \cdot \frac{1}{K} \sum_{k=1}^K \nabla \mathcal{L}_{\mathtt{Alg}}(\theta_{\mathrm{F}}^t; \tau_k^{(t)}, \mathcal{P}_{\mathrm{B}}(\cdot|\cdot, \theta_{\mathrm{B}}^{t+1}))$;
8: **end for**

---

**Convergence of the method**    Next, we show why this method indeed solves the GFlowNet learning problem. First, we introduce a *non-stationary soft reinforcement learning problem* of minimizing the worst-case dynamic average regret

$$\overline{\mathfrak{R}}^T := \frac{1}{T} \sum_{t=1}^T V_{\lambda=1}^{\mathcal{P}_{\mathrm{F}}^\star}(s_0; r^t) - V_{\lambda=1}^{\mathcal{P}_{\mathrm{F}}^t}(s_0; r^t)\,, \tag{9}$$

where $\{r^t\}_{t \in [T]}$ is a sequence of reward functions, and $r^t$ is revealed to a learner before selecting a policy $\mathcal{P}_{\mathrm{F}}^t$. Following Zahavy et al. (2021), we conjecture that existing RL algorithms are adaptive to the setting of known but non-stationary reward sequences. The implementation of Sampler player of `EntGame` algorithm by Tiapkin et al. (2023) is an example of such a regret minimization algorithm. Additionally, we notice that the optimization of dynamic regret is well-studied in the online learning literature, even in a more challenging setting of revealing the corresponding reward function *after* playing a policy (Zinkevich, 2003; Besbes et al., 2015).

Next, we provide the convergence result for our two-step procedure, using a stability argument for the first step. The proof is given in Appendix A.1.

**Theorem 3.1.** *Assume that (1) the backward updates are stable, i.e., $\sup_{t \geq 0} \|\mathcal{P}_{\mathrm{B}}^T - \mathcal{P}_{\mathrm{B}}^{T+t}\|_1 \to 0$ as $T \to \infty$, and (2) the forward updates follow a non-stationary regret minimization algorithm, i.e., $\overline{\mathfrak{R}}^T \to 0$ as $T \to \infty$. Then, there exists a proper GFlowNet forward policy $\mathcal{P}_{\mathrm{F}}^\star$ that induces the marginal distribution $\mathcal{R}(x)/Z$ over terminal states, such that $\frac{1}{T} \sum_{t=1}^T \mathsf{P}_{\mathcal{T}}^{\mathcal{P}_{\mathrm{F}}^t} \to \mathsf{P}_{\mathcal{T}}^{\mathcal{P}_{\mathrm{F}}^\star}$ as $T \to \infty$.*

During numerical experiments, we observed that enforcing stability in backward updates, particularly by using a decaying learning rate and a target network, significantly improves convergence in practice. Furthermore, as the theorem shows, this stability is essential for theoretical convergence. We discuss stability techniques we utilize alongside our algorithm in Appendix A.2.

## 4 EXPERIMENTS

We conduct our experimental evaluation on hypergrid (Bengio et al., 2021) and bit sequence (Malkin et al., 2022) environments, as well as on two molecule design environments: sEH (Bengio et al., 2021) and QM9 (Jain et al., 2023). Additional experimental details and hyperparameter choices are provided in Appendix A.3.

We evaluate four GFlowNet training methods: `MunchausenDQN` (based on the framework of Tiapkin et al. (2024)), `DB` (Bengio et al., 2023), `TB` (Malkin et al., 2022), and `SubTB` (Madan et al., 2023), which we refer to as *GFlowNet algorithms* (denoted as $\mathcal{L}_{\mathtt{Alg}}$ in the previous section). On hypergrids, we also include results for `SoftDQN` (Tiapkin et al., 2024). Alongside these algorithms, we consider five strategies for learning or selecting the backward policy:

- our approach (`TLM`);
- fixed uniform backward policy (`uniform`);
- simultaneously learning the backward policy with $\mathcal{P}_{\mathrm{F}}$ using the same objective (`naive`);
- maximum entropy backward policy (`maxent`) (Mohammadpour et al., 2024);
- pessimistic backward policy (`pessimistic`) (Jang et al., 2024).

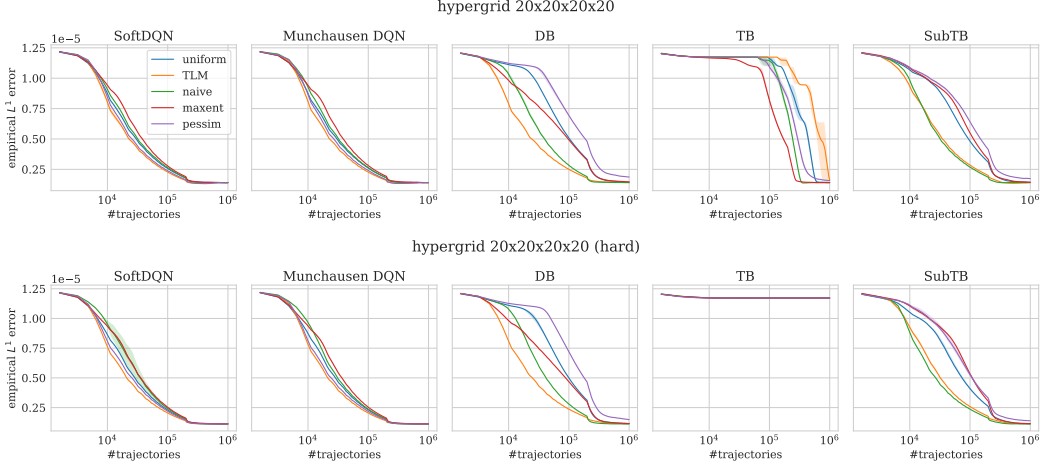

Figure 1: $L^1$ distance between target and empirical sample distributions over the course of training on the standard (**top row**) and hard (**bottom row**) hypergrid environments for each method. *Lower values indicate better performance.*

We refer to these strategies as *backward approaches*. In this section, we denote the distribution induced by a forward policy $\mathcal{P}_F$ over terminal states as $P_\theta(x)$ for $x \in \mathcal{X}$, representing the probability of sampling $x$ from our GFlowNet.

## 4.1 HYPERGRID

We start experiments with synthetic hypergrid environments introduced by Bengio et al. (2021). These environments are sufficiently small to compute target distribution in the closed form, allowing us to directly examine the convergence of $P_\theta(x)$ to $\mathcal{R}(x)/Z$.

The environment is a $d$-dimensional hypercube with a side length equal to $H$. The state space is represented as $d$-dimensional vectors $(s_1, \ldots, s_d)^\top \in \{0, \ldots, H-1\}^d$ with the initial state being $(0, \ldots, 0)^\top$. For each state $(s_1, \ldots, s_{d-1})$, there are at most $d+1$ actions. The first action always corresponds to an exit action that transfers the state to its terminal copy, and the rest of $d$ actions correspond to incrementing one coordinate by 1 without leaving the grid. The number of terminal states here is $|\mathcal{X}| = H^d$. There are $2^d$ regions with high rewards near the corners of the grid, while states outside have much lower rewards. The exact expression for the rewards is given in Appendix A.3.2.

We explore environments with the reward parameters taken from Malkin et al. (2022), referred to as "standard case", and with the reward parameters from Madan et al. (2023), referred to as a "hard case". In the second case, background rewards are lower, which makes mode exploration more challenging. We conduct experiments on a 4-dimensional hypercube with a side length of 20. As an evaluation metric, we use $L^1$ distance between the true reward distribution and the empirical distribution of the last $2 \cdot 10^5$ terminal states sampled during training.

Figure 1 presents the results. For `SoftDQN`, `MunchausenDQN`, and `DB`, `TLM` shows the fastest convergence for both "standard" and "hard" reward designs. For the `SubTB` algorithm, `TLM` shows similar performance to `naive` and outperforms `uniform`, `maxent` and `pessimistic`. `TB` is known to have difficulties in this environment (Madan et al., 2023), all approaches fail to converge under the "hard" reward design. At the same time, with the "standard" one, `maxent` backward shows the best convergence. An important note is that our results reproduce the findings of Tiapkin et al. (2024): for `SoftDQN` and `MunchausenDQN` training with `uniform` backward converges faster than with `naive` algorithm, while `TLM` shows stable improvement over `uniform`. The results and the ranking of algorithms are almost the same for `SoftDQN` and `MunchausenDQN`, so we leave only `MunchausenDQN` out of two for further experiments.

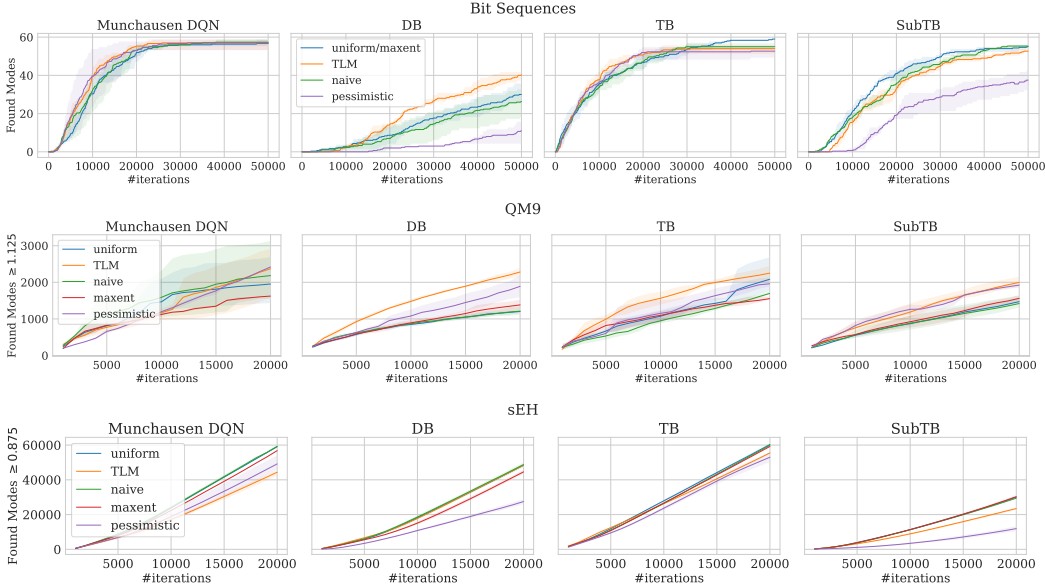

Figure 2: **Top row:** Bit Sequences, the number of discovered modes out of a total of 60 modes for different methods. **Center row:** QM9, the number of Tanimoto-separated modes with reward higher or equal to $1.125$ for different methods. **Bottom row:** sEH, the number of Tanimoto-separated modes with reward higher or equal to $0.875$ for different methods. *Higher values indicate better performance.* For each pair of a GFlowNet algorithm and a backward approach, the results are presented for the best learning rate chosen in terms of the total number of discovered modes.

## 4.2 BIT SEQUENCES

In this section, we consider the bit sequence generation task introduced by Malkin et al. (2022). Following the experimental setup of Tiapkin et al. (2024), we modify the state and action spaces to create a non-tree DAG structure, similar to the approach introduced in Zhang et al. (2022).

This task is to generate binary sequences of a fixed length $n$, using a vocabulary of $k$-bit blocks. The state space of this environment corresponds to sequences of $n/k$ words, and each word in these sequences is either an empty word $\oslash$ or one of $2^k$ possible $k$-bit words. The initial state $s_0$ corresponds to a sequence of empty words. The possible actions in each state are replacing an empty word $\oslash$ with one of $2^k$ non-empty words in the vocabulary. The set of terminal states $\mathcal{X}$ consists of sequences without empty words and corresponds to binary strings of length $n$. The reward function is defined as $\mathcal{R}(x) = \exp(-2 \cdot \min_{x' \in \mathcal{M}} d(x, x'))$, where $\mathcal{M}$ is a set of modes and $d$ is Hamming distance. We fix $n = 120$ and $k = 8$ for our experiments. The terminal state space size is $|\mathcal{X}| = 2^{120}$. Importantly, for this environment, the `uniform` backward coincides with `maxent`, see Proposition 1 of Zhang et al. (2022) and Remark 3 of Theorem 3 of Mohammadpour et al. (2024).

To evaluate the performance, we use the same metrics as in Malkin et al. (2022) and Tiapkin et al. (2024): the number of modes found during training (number of sequences from $\mathcal{M}$ for which a terminal state within a distance of 30 has been sampled) and Spearman correlation on the test set between $\mathcal{R}(x)$ and an estimate of $P_\theta$. Since computing the exact probability of sampling a terminal state is intractable due to a large number of paths leading to it, we use a Monte Carlo estimate following the approach of Zhang et al. (2022); see also Appendix A.3. We train all models with various choices of the learning rate, treating it as a hyperparameter, and provide the results depending on its value, similarly to Madan et al. (2023).

Figure 2 shows the number of modes for different GFlowNet algorithms and backward approaches found over the course of training. We observe that `TLM` shows a significant improvement for `DB` and a minor one for `MunchausenDQN` in comparison to other backward approaches, where in the latter case all backward approaches find all 60 modes. `TB` and `SubTB` also find almost all modes, and

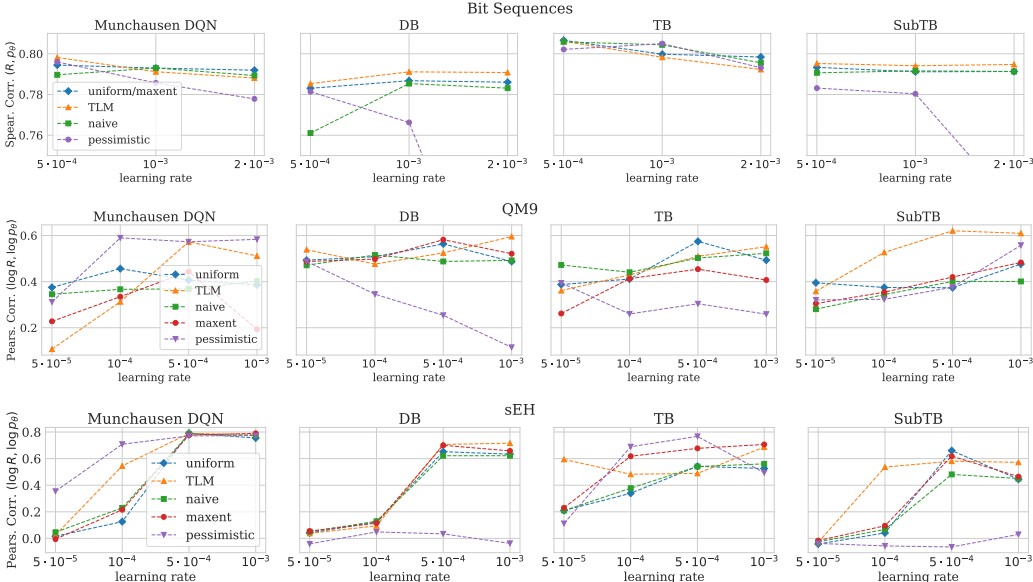

Figure 3: **Top row:** Bit Sequences, Spearman correlation between $\mathcal{R}$ and $P_\theta$ on a test set for different methods and varying learning rate $\in \{5 \cdot 10^{-4}, 10^{-3}, 2 \cdot 10^{-3}\}$. **Center row:** QM9, Pearson correlation between $\log \mathcal{R}$ and $\log P_\theta$ on the fixed subset of the QM9 dataset (Ramakrishnan et al., 2014) for different methods and varying learning rate $\in \{5 \cdot 10^{-5}, 10^{-4}, 5 \cdot 10^{-4}, 10^{-3}\}$. **Bottom row:** sEH, Pearson correlation between $\log \mathcal{R}$ and $\log P_\theta$ on the test set from Bengio et al. (2021) for different methods and varying learning rate $\in \{5 \cdot 10^{-5}, 10^{-4}, 5 \cdot 10^{-4}, 10^{-3}\}$. *Higher values indicate better performance*. We note here that `pessimistic` backward policy can be very sensitive to the choice of learning rate.

`TLM` does not affect the results much in comparison to `uniform` and `naive`, while outperforming `pessimistic` in case of `SubTB`. Full plots for modes across varying learning rates are presented in Figure 5 in Appendix. Figure 3 (top) presents Spearman correlation between $\mathcal{R}$ and $P_\theta$ on the test set for the same GFlowNet algorithms and varying learning rates. `TLM` shows better or similar performance to the baselines across all GFlowNet algorithms if the optimal learning rate is chosen. Moreover, for `DB` and `SubTB`, `TLM` shows steady improvement over the baselines for all learning rates.

## 4.3 MOLECULE DESIGN, sEH AND QM9

Our final experiments are carried out on molecule design tasks of sEH (Bengio et al., 2021) and QM9 (Jain et al., 2023). In both tasks, the goal is to generate molecular graphs, with reward emphasizing some desirable property. For both problems, we use pre-trained reward proxy neural networks. For the sEH task, the model is trained to predict the binding energy of a molecule to a particular protein target (soluble epoxide hydrolase) (Bengio et al., 2021). For the QM9 task, the proxy is trained on the QM9 dataset (Ramakrishnan et al., 2014) to predict the HOMO-LUMO gap (Zhang et al., 2020).

For the sEH task, we follow the framework proposed by Jin et al. (2020) and generate molecules using a predefined vocabulary of fragments. We use the same 72 fragments as in the seminal work of Bengio et al. (2021). It is essential to mention that these fragments are explicitly selected for the sEH task to simplify high-quality object generation. The states are represented as trees of fragments. The actions correspond to choosing a new fragment and then choosing an atom to which the fragment will be attached. There is also a special stop action that moves the state to its terminal copy and stops the generation process.

For the QM9 task, molecules are generated atom-by-atom and bond-by-bond. Every state is a connected graph, and actions either add a new node and edge or set an attribute on an edge. Thus, the graph-building environment is much more expressive than the fragment-based tree-building

environment, but it results in a more complex generation task and can lead to construction of invalid molecules. It is worth mentioning that there exists a simpler fragment-based version of this environment in GFlowNet literature (Shen et al., 2023; Chen & Mauch, 2023), while we consider the more complex atom-by-atom setup from Mohammadpour et al. (2024).

We use the same evaluation metrics for both tasks as proposed in previous works (Bengio et al., 2021; Madan et al., 2023; Tiapkin et al., 2024). We track the number of Tanimoto-separated modes above a certain reward threshold captured over the course of training, and Pearson correlation on the test set between $\log \mathcal{R}(x)$ and $\log P_\theta(x)$. For sEH task we use the same test set as in Bengio et al. (2021), and for QM9 we use a subset of the dataset introduced in Ramakrishnan et al. (2014). We train all models with various choices of the learning rate, treating it as a hyperparameter, and provide the results depending on its value, similarly to Madan et al. (2023).

Figure 2 (center and bottom) shows the number of modes for different GFlowNet algorithms and backward approaches found over the course of training. `TLM` speeds up mode discovery on QM9 when utilized alongside `DB` and `DB` and performs on par with the best baseline backward approaches for `MunchausenDQN` and `SubTB`, but shows similar or worse performance when compared to other backward approaches on sEH. However, we note that on sEH no backward approach shows significant improvement over `uniform` in terms of mode discovery. Full plots for modes across varying learning rates are presented in Figure 6 in Appendix. It is worth noting that on the QM9 task, `TLM` shows robust improvement over the baselines in the majority of configurations. Figure 3 (center and bottom) shows Pearson correlation between $\log \mathcal{R}$ and $\log P_\theta$ estimate measured on the test set for various learning rates. `TLM` results in better correlations when paired with `SubTB`, shows comaprable results with the best baselines when paired with `DB` and `TB`, and falls behind `pessimistic` when paired with `MunchausenDQN` while outperforming other backward approaches. However, as Figure 3 indicates, `pessimistic` backward policy can be very sensitive to the choice of the learning rate and the GFlowNet algorithm that is used for forward policy training.

## 4.4 Discussion

From the plots above, one can see that across all GFlowNet algorithms (forward policy training objectives), `TLM` generally shows performance that is better or comparable to other backward approaches. The sole exception is the number of discovered modes in the sEH environment, where `TLM` can fall behind other backward approaches. To explain this shortcoming, we hypothesize that `TLM` is more beneficial in environments with less structure. This is supported by the major improvements to mode exploration that it obtains on QM9, while sometimes even degrading the same metric on sEH. Indeed, molecules in the sEH task are constructed from the predefined set of blocks, while in the QM9 task, they are created from atoms. This manually predefined set adds a lot of structure into the environment, in which one creates a junction tree from these blocks. In comparison, the process of creating an arbitrary graph of atoms imposes less structure and can even lead to construction of invalid molecules in certain cases. We suppose that the strong methodological bias is a possible reason why it is of little utility to consider non-trivial backward approaches in the sEH task and why the `uniform` backward approach often has the best or at least comparable performance according to Figure 6. This is exactly the hypothesis that was initially put forward by Mohammadpour et al. (2024), and our results align with it well.

## 5 Conclusion

In this work, we propose a new method for backward policy optimization that enhances mode exploration and accelerates convergence in complex GFlowNet environments. `TLM` represents the first principled method for learning a backward policy in soft reinforcement learning-based GFlowNet algorithms, such as `SoftDQN` and `MunchausenDQN`. We provide an extensive experimental evaluation, demonstrating benefits of `TLM` when it is paired with various forward policy training methods, and analyze its shortcomings, arguing that our results support the hypothesis of Mohammadpour et al. (2024) about benefits of backward policy optimization in environments with less structure.

A promising further work direction is using backward policy for exploration as proposed in Kim et al. (2023) and He et al. (2024b). Indeed, the ability to sample trajectories starting from high-reward terminal states via $\mathcal{P}_B$ provides an opportunity to improve mode exploration. We expect that combining such methods with `TLM`-like approaches will additionally improve their performance.

ACKNOWLEDGMENTS

The authors are grateful to Dmitry Vetrov for valuable discussions and feedback. The paper was prepared within the framework of the HSE University Basic Research Program. This research was supported in part through computational resources of HPC facilities at HSE University (Kostenetskiy et al., 2021). The work of Daniil Tiapkin was supported by the Paris Île-de-France Région in the framework of DIM AI4IDF.

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

# A   APPENDIX

## A.1   OMITTED PROOFS

*Proof of Theorem 3.1.* From the stability of backward updates, the Cauchy criterion implies that there is $\mathcal{P}_B^\star \in \Pi_B$ such that $\mathcal{P}_B^T \to \mathcal{P}_B^\star$. At the same time, by the choice or rewards $r^t = r^{\mathcal{P}_B^t}$, Proposition 1 by Tiapkin et al. (2024) and joint convexity of KL-divergence

$$\overline{\mathfrak{R}}^T = \frac{1}{T}\sum_{t=1}^{T}\mathrm{KL}\left(\mathsf{P}_{\mathcal{T}}^{\mathcal{P}_F^t}\middle\|\mathsf{P}_{\mathcal{T}}^{\mathcal{P}_B^t}\right) \geq \mathrm{KL}\left(\frac{1}{T}\sum_{t=1}^{T}\mathsf{P}_{\mathcal{T}}^{\mathcal{P}_F^t}\middle\|\frac{1}{T}\sum_{t=1}^{T}\mathsf{P}_{\mathcal{T}}^{\mathcal{P}_B^t}\right).$$

Notice that a mapping $\mathcal{P}_B \mapsto \mathsf{P}_{\mathcal{T}}^{\mathcal{P}_B}$ is continuous, thus $\mathsf{P}_{\mathcal{T}}^{\mathcal{P}_B^T} \to \mathsf{P}_{\mathcal{T}}^{\mathcal{P}_B^\star}$, and, as a result, averages of $\mathsf{P}_{\mathcal{T}}^{\mathcal{P}_B^t}$ also converge to $\mathsf{P}_{\mathcal{T}}^{\mathcal{P}_B^\star}$. Finally, applying Pinkser's inequality, we have

$$\left\|\tfrac{1}{T}\sum_{t=1}^{T}\mathsf{P}_{\mathcal{T}}^{\mathcal{P}_F^t} - \mathsf{P}_{\mathcal{T}}^{\mathcal{P}_B^\star}\right\|_1 \leq \sqrt{2\overline{\mathfrak{R}}^T} + \|\tfrac{1}{T}\sum_{t=1}^{T}\mathsf{P}_{\mathcal{T}}^{\mathcal{P}_B^t} - \mathsf{P}_{\mathcal{T}}^{\mathcal{P}_B^\star}\|_1.$$

The right-hand side of the inequality tends to zero as $T \to +\infty$, thus $\frac{1}{T}\sum_{t=1}^{T}\mathsf{P}_{\mathcal{T}}^{\mathcal{P}_F^t} \to \mathsf{P}_{\mathcal{T}}^{\mathcal{P}_B^\star}$. Finally, since for any $\mathcal{P}_B^\star$ there is $\mathcal{P}_F^\star$ such that $\mathsf{P}_{\mathcal{T}}^{\mathcal{P}_B^\star} = \mathsf{P}_{\mathcal{T}}^{\mathcal{P}_F^\star}$, we conclude the statement. □

## A.2   STABILITY TECHNIQUES

In this section, we highlight important practical techniques and design choices motivated by Theorem 3.1 that we use alongside our `TLM` algorithm to enforce stability into the training of $\mathcal{P}_B$.

First, we found it beneficial to either use a lower learning rate for the backward policy or decay it over the course of training (see the next sections for detailed descriptions).

Second, akin to how the Deep Q-Network algorithm (Mnih et al., 2015) utilizes a target network to estimate the value of the next state, we utilize target networks for the backward policy when calculating the loss for the forward policy. For example, (3) transforms into

$$\mathcal{L}_{\mathrm{SubTB}}(\theta;\tau) = \sum_{0\leq j<k\leq n_\tau} w_{jk}\left(\log\frac{F_\theta(s_j)\prod_{t=j+1}^{k}\mathcal{P}_F(s_t|s_{t-1},\theta)}{F_\theta(s_k)\prod_{t=j+1}^{k}\mathcal{P}_B(s_{t-1}|s_t,\bar\theta)}\right)^2 \tag{10}$$

where the parameters $\bar\theta$ of $\mathcal{P}_B(s_{t-1}|s_t,\bar\theta)$ are updated via exponential moving average (EMA) of the online parameters $\theta$ of $\mathcal{P}_B(s_{t-1}|s_t,\theta)$. So the loss for the backward policy $\mathcal{L}_{\mathrm{TLM}}$ is computed using an online backward policy $\mathcal{P}_B(s_{t-1}|s_t,\theta)$, and the loss for the forward policy $\mathcal{L}_{\mathrm{Alg}}$ is computed using a target backward policy $\mathcal{P}_B(s_{t-1}|s_t,\bar\theta)$, which is frozen during the gradient update of $\mathcal{P}_F$.

Finally, we find it helpful to initialize $\mathcal{P}_B$ to the uniform distribution at the beginning of training, which is done by zero-initialization of the last linear layer weight and bias.

We ablate the impact of the proposed techniques on QM9, where we try to turn off each of the three separately. Results are presented in Figure 4. We observe that using a target model and a lower learning rate is crucial, whereas disabling uniform initialization increases variance and shows slightly worse results. For this experiment, we choose `DB` as the base algorithm because `TLM` overall shows the most significant impact when applied with it compared to `TB`, `SubTB`, and `MunchausenDQN`.

## A.3   EXPERIMENTAL DETAILS

We utilize PyTorch (Paszke et al., 2019) in our experiments. For hypergrid and bit sequence environments, we base our implementation upon the published code of Tiapkin et al. (2024). For molecule design experiments, our implementations are based on the open source library by Recursion Pharmaceuticals.[1] In all our experiments, $\mathcal{P}_F$ and $\mathcal{P}_B$ share the same neural network backbone, predicting logits via two separate linear heads. For all experiments, mean and std values are computed over three random seeds.

---

[1] https://github.com/recursionpharma/gflownet

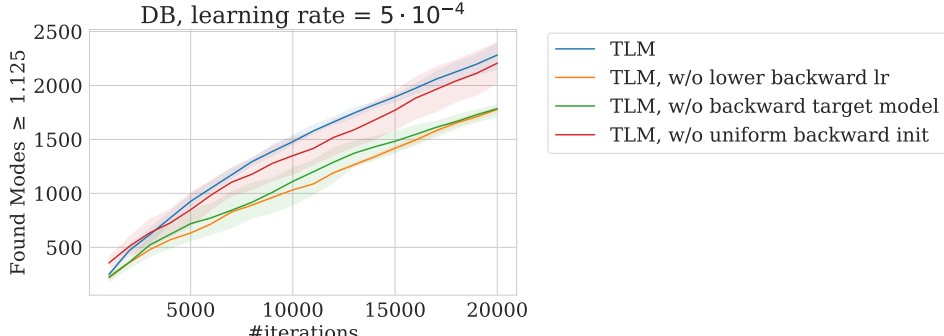

Figure 4: Ablation study of stability techniques on QM9. The number of Tanimoto-separated modes with a reward at least $1.125$ is shown. As a base algorithm, we use `DB` with a learning rate of $5 \cdot 10^{-4}$.

An important note is that we do one stochastic gradient update over the replay buffer for `pessimistic` backward policy approach (Jang et al., 2024) at each training iteration to allow for a fair comparison with other methods. Jang et al. (2024) perform 8 stochastic gradient updates for `pessimistic` $\mathcal{P}_B$ at each training iteration in the experiments in their paper, which results in a discrepancy in the number of gradient steps when comparing the approach to other methods.

### A.3.1 HYPERGRID

The reward at a terminal state $s$ with coordinates $(s^1, \dots, s^D)$ is defined as

$$
\mathcal{R}(s) = R_0 + R_1 \times \prod_{i=1}^{D} \mathbb{I}\left[0.25 < \left| \frac{s^i}{H-1} - 0.5 \right|\right] + R_2 \times \prod_{i=1}^{D} \mathbb{I}\left[0.3 < \left| \frac{s^i}{H-1} - 0.5 \right| < 0.4\right].
$$

Standard reward uses parameters $(R_0 = 10^{-3}, R_1 = 0.5, R_2 = 2.0)$ and hard reward uses $(R_0 = 10^{-4}, R_1 = 1.0, R_2 = 3.0)$, taken from Bengio et al. (2021) and Madan et al. (2023) respectively.

All models are parameterized using an MLP with 2 hidden layers and 256 hidden units. We use the Adam optimizer with a learning rate of $10^{-3}$ and a batch size of 16 trajectories. For `SubTB`, we set $\lambda = 0.9$, following Madan et al. (2023). For `SoftDQN` and `MunchausenDQN`, we use a prioritized replay buffer (Schaul et al., 2016) and adopt the same hyperparameters as in Tiapkin et al. (2024). We sample 256 transitions from the buffer to compute the loss for these methods.

For the `TLM` backward policy, we use the same initial learning rate of $10^{-3}$ with an exponential scheduler, tuning $\gamma$ from $\{0.999, 0.9999\}$. The backward policy target network for `TLM` uses soft updates with a parameter $\tau = 0.25$ (Silver et al., 2014). Since the environment is small, we precompute $\log n(s)$ for all states, allowing us to obtain the `maxent` backward exactly. For `pessimistic` backward policy we store trajectories from the last 20 training iterations in the replay buffer and sample 16 trajectories to compute the loss.

Hypergrid experiments were performed on CPUs. Table 1 summarizes the chosen hyperparameters.

### A.3.2 BIT SEQUENCES

The set of modes $M$ is defined as in Malkin et al. (2022), and we choose the same size, $|M| = 60$. We set $H = \{'00000000','11111111','11110000','00001111','00111100'\}$. Each sequence in $M$ is generated by randomly selecting $n/8$ elements from $H$ with replacement and then concatenating them. The test set for evaluating reward correlations is generated by taking a mode and flipping $i$ random bits in it, where this is repeated for every mode and for each $0 \leq i < n$.

We utilize the same Monte Carlo estimate for $P_\theta$ as presented in Tiapkin et al. (2024) with $N = 10$:

$$
P_\theta(x) = \mathbb{E}_{\mathcal{P}_B(\tau|x)}\left[\frac{\mathcal{P}_F(\tau \mid \theta)}{\mathcal{P}_B(\tau \mid x)}\right] \approx \frac{1}{N}\sum_{i=1}^{N}\frac{\mathcal{P}_F(\tau^i \mid \theta)}{\mathcal{P}_B(\tau^i \mid x)}, \quad \tau^i \sim \mathcal{P}_B(\tau \mid x).
$$

Table 1: Hyperparameter choices for hypergrid experiments.

| Hyperparatemeter | Value |
|---|---|
| Training trajectories | $10^6$ |
| Learning rate | $10^{-3}$ |
| $Z$ learning rate for `TB` | $10^{-1}$ |
| Adam optimizer $\beta, \varepsilon, \lambda$ | $(0.9, 0.999), 10^{-8}, 0$ |
| Number of hidden layers | 2 |
| Hidden embedding size | 256 |
| `SubTB` $\lambda$ | 0.9 |
| `MunchausenDQN` $\alpha$ | 0.15 |
| `MunchausenDQN` $l_0$ | $-100$ |
| Prioritized RB size | $10^5$ |
| Prioritized RB $\alpha, \beta$ | 0.5, 0.0 |
| Prioritized RB batch size | 256 |
| `pessimistic` buffer size | $20 \times 16$ |
| `TLM` backward learning rate | $10^{-3}$ |
| `TLM` backward lr decay $\gamma$ | $\{0.999, 0.9999\}$ |
| `TLM` backward target model $\tau$ | 0.25 |

Notice that any valid $\mathcal{P}_B$ can be used here, but for each model, we take the $\mathcal{P}_B$ that was fixed/trained alongside the corresponding $\mathcal{P}_F$ since such a choice will lead to a lower estimate variance. However, we note that the metric is still very noisy, so for each training run, we compute the metric for all model checkpoints (done every 2000 iterations) and use the maximum value.

All models are parameterized as Transformers Vaswani et al. (2017) with 3 hidden layers, 8 attention heads, and a hidden dimension of 64. Each model is trained for 50,000 iterations and a batch size of 16 with Adam optimizer. We provide results for learning rates from $\{5 \times 10^{-4}, 10^{-3}, 2 \times 10^{-3}\}$. For `SubTB` we use $\lambda = 0.9$. For `MunchausenDQN`, we use a prioritized replay buffer (Schaul et al., 2016) and take the same hyperparameters as in Tiapkin et al. (2024). To compute the loss, we sample 256 transitions from the buffer for `MunchausenDQN`.

For the `TLM` backward policy, we use the same initial learning rate as for the forward policy and utilize an exponential scheduler, where $\gamma$ is tuned from $\{0.9997, 0.9999\}$. The backward policy target network for `TLM` uses soft updates with a parameter $\tau = 0.1$ (Silver et al., 2014). For `pessimistic` backward policy we store trajectories from the last 20 training iterations in the replay buffer and sample 16 trajectories to compute the loss.

To closely follow the setting of previous works (Malkin et al., 2022; Madan et al., 2023; Tiapkin et al., 2024), we use $\varepsilon$-uniform exploration with $\varepsilon = 10^{-3}$. We note that this can introduce a small bias into the gradient estimate of $\overline{\nabla}_\theta \mathcal{L}_{\text{TLM}}(\theta_B^t; \tau)$ since $\tau$ will not be sampled exactly from $\mathcal{P}_F$.

Each bit sequence experiment was performed on a single NVIDIA V100 GPU. Table 2 summarizes the chosen hyperparameters.

### A.3.3 MOLECULES

For sEH, we use the test set from Bengio et al. (2021). For QM9, we select a subset of 773 molecules from the QM9 dataset (Ramakrishnan et al., 2014) containing between 3 and 8 atoms. The subset is constructed to ensure an approximately equal representation of different molecule sizes. To compute correlation, we use the same Monte Carlo estimate as in the bit sequence task. It is important to note that Mohammadpour et al. (2024) used a different evaluation approach, computing correlation on sampled molecules rather than on a fixed dataset.

In the sEH task, rewards are divided by a constant of 8, and the reward exponent is set to 10. For QM9, we subtract the 95th percentile from all rewards, resulting in the majority of rewards being distributed between 0 and 1, with 5% of molecules having a reward greater than 1. The reward exponent is also set to 10. We set $\mathcal{R}(x) = \exp(-75.0)$ for invalid molecules $x$ in QM9. We track the number of Tanimoto-separated modes as described in Bengio et al. (2023), using a Tanimoto

Table 2: Hyperparameter choices for bit sequence experiments.

| Hyperparatemeter | Value |
|---|---|
| Training iterations | $5 \times 10^4$ |
| Learning rate | $\{5 \times 10^{-4}, 10^{-3}, 2 \times 10^{-3}\}$ |
| $Z$ learning rate for `TB` | $10^{-1}$ |
| Adam optimizer $\beta, \varepsilon, \lambda$ | $(0.9, 0.999), 10^{-8}, 10^{-5}$ |
| $\varepsilon$-uniform exploration | $10^{-3}$ |
| Number of transformer layers | 3 |
| Hidden embedding size | 64 |
| Dropout | 0.1 |
| `SubTB` $\lambda$ | 0.9 |
| `MunchausenDQN` $\alpha$ | 0.15 |
| `MunchausenDQN` $l_0$ | $-100$ |
| `MunchausenDQN` leaf coefficient | 5 |
| Prioritized RB size | $10^5$ |
| Prioritized RB $\alpha, \beta$ | $0.9, 0.1$ |
| Prioritized RB batch size | 256 |
| `pessimistic` buffer size | $20 \times 16$ |
| `TLM` backward learning rate | learning rate |
| `TLM` backward lr decay $\gamma$ | 0.9999 |
| `TLM` backward target model $\tau$ | 0.1 |

similarity threshold of 0.7. After normalization, the reward thresholds are $0.875$ and $1.125$ for sEH and QM9, respectively.

We use the graph transformer architecture from Jain et al. (2023) with 8 layers and 256 embedding dimensions for both tasks. Each model is trained for 20,000 iterations using the Adam optimizer. We present results for learning rates in the set $\{5 \times 10^{-4}, 10^{-4}, 5 \times 10^{-4}, 10^{-3}\}$. The learning rate for Z is fixed at $10^{-3}$ for all experiments. We apply exponential schedulers to all learning rates at each training step and refer readers to the hyperparameter table for the exact decay values. The batch sizes are 256 for sEH and 128 for QM9. For `SubTB`, we set $\lambda = 1.0$, following Madan et al. (2023). For `MunchausenDQN`, we do not use a replay buffer, training the model on-policy while using the same other hyperparameters as in Tiapkin et al. (2024).

For `TLM`, the learning rate for $\mathcal{P}_B$ is initially set to be 10 times smaller than that for $\mathcal{P}_F$, while other approaches use the same learning rate for $\mathcal{P}_B$ as for $\mathcal{P}_F$. The backward policy target network for `TLM` uses soft updates with a parameter $\tau = 0.05$ (Silver et al., 2014). To learn $\log n(s)$ for the `maxent` backward policy, we follow the approach from Mohammadpour et al. (2024), implemented in the Recursion Pharmaceuticals GFlowNet library. For `pessimistic` backward policy we store trajectories from the last 20 training iterations in the replay buffer and sample 256 trajectories to compute the loss for sEH and 128 for QM9 (the same as batch size for these environments).

In line with previous works (Malkin et al., 2022; Madan et al., 2023; Tiapkin et al., 2024), we use $\varepsilon$-uniform exploration with $\varepsilon = 0.05$. To account for the bias introduced into the gradient estimate of $\nabla_\theta \mathcal{L}_{\text{TLM}}(\theta_B^t; \tau)$, we linearly anneal $\varepsilon$ to zero over the course of training. This annealing approach is applied to all GFlowNet algorithms and backward methods to ensure a fair comparison in terms of the number of discovered modes.

Each molecule generation experiment was conducted on a single NVIDIA A100 GPU. Table 3 summarizes the chosen hyperparameters.

Table 3: Hyperparameter choices for molecule experiments.

| Hyperparatemeter | sEH | QM9 |
|---|---|---|
| Reward exponent | 10 | |
| Batch size | 256 | 128 |
| Training iterations | $2 \times 10^4$ | |
| Learning rate | $\{5 \times 10^{-4}, 10^{-4}, 5 \times 10^{-4}, 10^{-3}\}$ | |
| Learning rate decay $\gamma$ | $2^{-1/20000}$ | |
| $Z$ learning rate for `TB` | $10^{-3}$ | |
| $Z$ learning rate for `TB` decay | $2^{-1/50000}$ | |
| Adam optimizer $\beta, \varepsilon, \lambda$ | $(0.9, 0.999), 10^{-8}, 10^{-8}$ | |
| $\varepsilon$-uniform exploration | 0.05 linearly annealed to 0.0 | |
| Number of transformer layers | 8 | |
| Hidden embedding size | 256 | |
| `SubTB` $\lambda$ | 1.0 | |
| `MunchausenDQN` $\alpha$ | 0.15 | |
| `MunchausenDQN` $l_0$ | $-500$ | |
| `MunchausenDQN` leaf coefficient | 10 | |
| pessimistic buffer size | $20 \times 256$ | $20 \times 128$ |
| TLM backward learning rate | $0.1\times$ learning rate | |
| TLM backward lr decay $\gamma$ | $2^{-1/5000}$ | $2^{-1/20000}$ |
| TLM backward target model $\tau$ | 0.05 | |

## A.4 FULL PLOTS

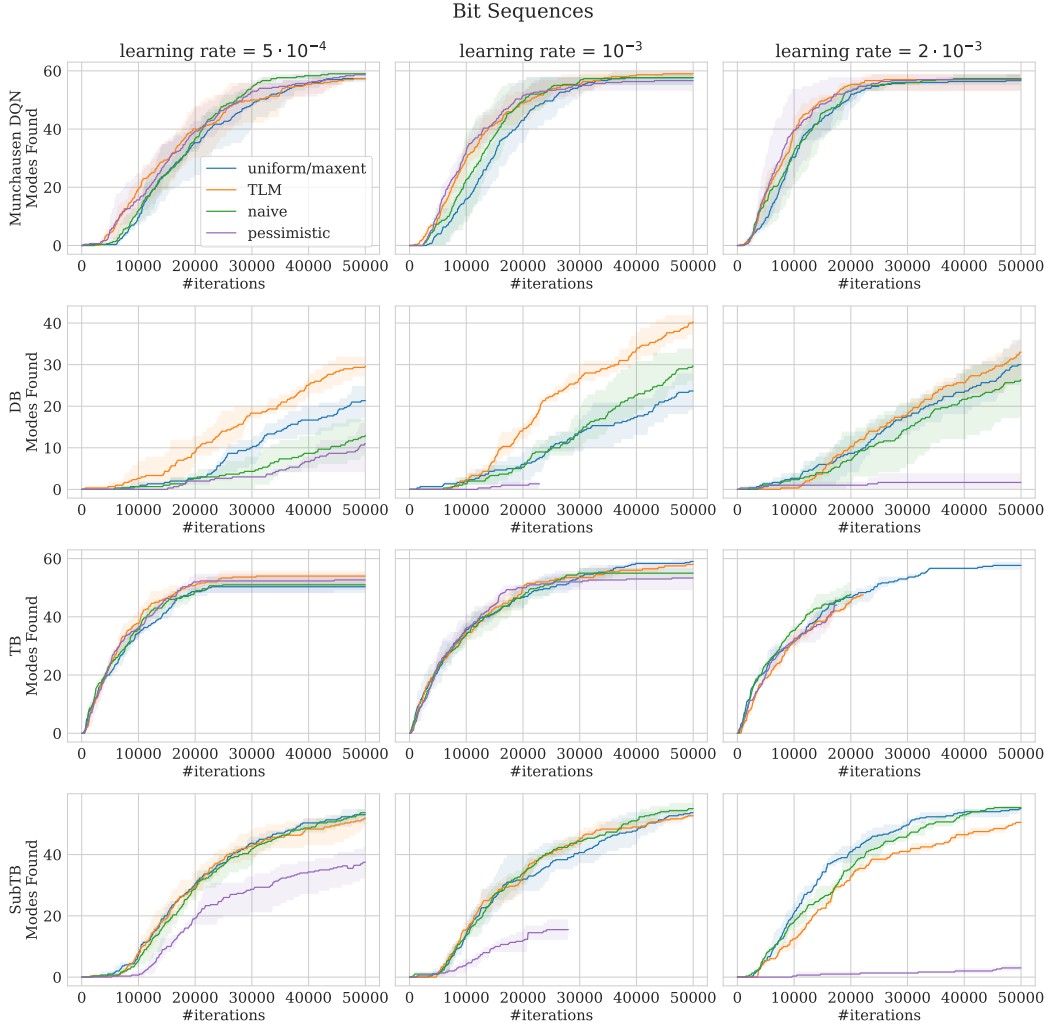

Figure 5: Bit Sequences, the number of modes discovered over the course of training for different methods and a learning rate $\in \{5 \cdot 10^{-4}, 10^{-3}, 2 \cdot 10^{-3}\}$. Some results for the learning rates of $10^{-3}$ and $2 \cdot 10^{-3}$ are not full because of exploding gradients at certain points in training.

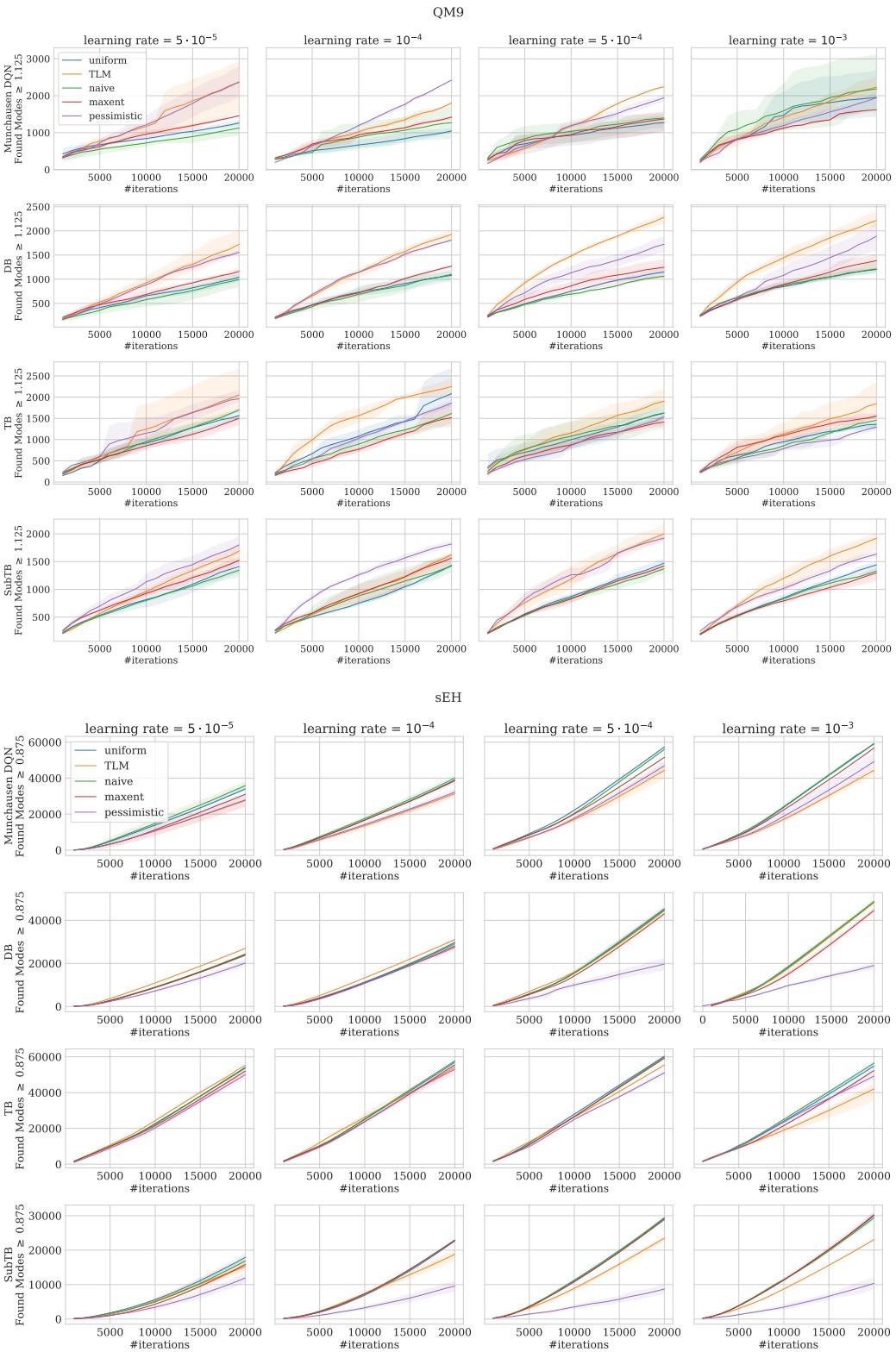

Figure 6: **Top row:** QM9, the number of Tanimoto-separated modes discovered over the course of training with reward higher or equal to $1.125$ for different methods and learning rate in $\in \{5 \cdot 10^{-5}, 10^{-4}, 5 \cdot 10^{-4}, 10^{-3}\}$. **Bottom row:** sEH, the number of Tanimoto-separated modes discovered over the course of training with reward higher or equal to $0.875$ for different methods and learning rate $\in \{5 \cdot 10^{-5}, 10^{-4}, 5 \cdot 10^{-4}, 10^{-3}\}$.

