# OpenReview forum: "Optimizing Backward Policies in GFlowNets via Trajectory Likelihood Maximization"
_ICLR.cc/2025/Conference — ICLR 2025 Poster_

### Official Review · Reviewer_EZ4M · 2024-10-21

**Soundness:** 3
**Presentation:** 3
**Contribution:** 2
**Rating:** 5
**Confidence:** 4

**Summary:**

This paper introduces a novel method called Trajectory Likelihood Maximization (TLM) for optimizing the backward policy in Generative Flow Networks (GFlowNets). GFlowNets are a class of generative models, and the TLM method accelerates convergence in complex tasks by directly maximizing the likelihood of trajectories. Additionally, it significantly enhances exploration capabilities in high-dimensional tasks.

**Strengths:**

1. The TLM method shares a strong theoretical connection with existing reinforcement learning approaches, such as entropy-regularized RL. It introduces a novel joint optimization framework that enhances the efficiency of learning the backward policy. These theoretical connections are validated through experimental results presented in the paper.

2. The method is grounded in the maximum likelihood estimation problem from reinforcement learning and proposes a unified approach for optimizing the backward policy. This provides a robust theoretical basis for addressing the longstanding challenge of backward policy optimization in GFlowNets.

3. The paper is written in a clear and accessible manner, particularly in its detailed explanations of complex theoretical concepts, making it easier for readers to comprehend.

**Weaknesses:**

1. The paper primarily focuses on the introduction and validation of the TLM method, but provides limited discussion on comparisons with other backward policy methods, such as those proposed by Jang et al. and Mohammadpour et al. Notably, the improvement in backward policy optimization is not significant in more structured tasks, such as the sEH task.

2. While the paper demonstrates the advantages of the TLM method in complex environments, it does not sufficiently explore scenarios where TLM might underperform, particularly in simpler or highly structured tasks.

3. Some of the experimental environments, such as the sEH task, are relatively simple. Although the TLM method performs well in these tasks, validation in more complex or realistic environments would better illustrate the practical value of the approach.

4. Although the paper emphasizes the TLM method, exploring combinations with other backward policy approaches, such as maximum entropy policies, could offer more innovative directions for future research.

5. The paper could further discuss the impact of dynamic reward structures on the TLM method, particularly in handling complex, non-stationary reward distributions.

**Questions:**

1. The paper highlights the advantages of the TLM method in complex environments, but its performance is suboptimal in simpler environments, such as the sEH task. Can the reasons behind this underperformance in simpler tasks be explored in more depth? Is it due to limitations of the model or characteristics inherent to the environment?

2. The paper notes that TLM struggles to handle non-stationary rewards effectively. Could a variant of TLM be designed specifically for dynamic reward structures to address this issue? How would such an adaptation impact TLM's performance in non-stationary reward environments?

3. Maximum entropy policies are beneficial for enhancing exploration. Could the TLM method be combined with maximum entropy policies to improve exploration capabilities in certain tasks? If so, how might these two approaches be effectively integrated for optimal collaboration?

4. Could the TLM method be integrated with other types of reinforcement learning algorithms, such as hard-constrained RL, to address different task requirements? Would such a combination further improve backward policy optimization and overall performance?

---

> ### Author Response · Authors · 2024-11-18
> **Author Response (Part 1)**
>
> We thank the reviewer for their valuable suggestions and are happy to provide further details in response to their questions.
>
> [Weakness 1] Firstly, we would like to respectfully disagree with the first stated weakness and note that we do directly compare in all experiments with the maxent backward approach proposed in [1] and discuss this approach in detail in our paper (Section 2.3). As for [2], we note that at the moment of the ICLR 2025 submission deadline, the paper [2] had not yet made its code publicly available, and the implementation details provided in the arXiv preprint were limited, making it challenging to reproduce the presented approach. After the ICLR 2025 submission deadline, the authors published the code for their method and experiments. Although it will be unfeasible to make a careful and faithful comparison during the time allocated for rebuttal, we are more than willing to include a comprehensive comparison to the approach of [2] in the camera-ready version of our paper.
>
> [Weaknesses 1-3, Question 1] Secondly, we note that sEH is a challenging molecule generation environment, introduced in the seminal work on GFlowNets [3] and used as a benchmark in many GFlowNet papers, with the number of states $\approx 10^{16}$ and number of actions from $100$ to $2000$ (see [3], Section 4.2). There exists a simplified version of the environment in GFlowNet literature with a reduced number of fragments and thus reduced state space, see, e.g. [5]. However, we emphasize that this is not the setting considered in our submission. While we highlight in our paper that sEH is a more structured environment than QM9, it does not mean it is simple. No method obtains the correlation with a reward higher than $0.75$ on sEH in the presented experiments, meaning that the environment is far from being solved exactly.
>
> However, we appreciate the reviewer’s comments on sEH task and the differences in TLM performance between different environments. We added a detailed discussion of the experimental results to the end of Section 4 to improve the clarity of our paper regarding these points. Concerning the poor performance on sEH, we argue that our experimental results align with the hypothesis by [1], which states that backward policy optimization is more beneficial in environments with less structure, like QM9. While for highly structured tasks like sEH, considering non-trivial backward approaches is of little utility and using a simple uniform backward policy generally shows strong performance. We refer the reviewer to the updated manuscript for a more detailed discussion.
>
> [Weakness 4, Question 3] We appreciate the question about combining our approach with maxent backward [1]. Unfortunately, the direct combination is impossible because there exists a unique backward policy that maximizes the trajectory entropy, as it was proven in [1], and the essence of their approach is to obtain such a policy exactly. Our method focuses on training the backward policy to better align with the current forward policy in terms of trajectory distributions. However, a possible idea could be to add the entropy of the backward policy as a regularizer to our objective, promoting both diversity and alignment to the forward policy. We thank the reviewer for this valuable suggestion and would like to consider it as an interesting future research direction.
>
> [Weakness 5, Question 2] Regarding the non-stationary reward problem, we would like to highlight that the TLM method introduces the non-stationary rewards to the RL formulation of the GFlowNet training problem, as does any backward policy optimization method. Indeed, the RL formulation of GFlowNet training considers the backward probabilities as a part of the intermediate reward functions. In our work, it was the only context in which we have mentioned the non-stationary RL problem, and we experimentally demonstrated that TLM handles it well.  At the same time, the setting of the non-stationary GFlowNet reward, which is defined only for terminal states, was never explored in the literature directly; however, it naturally appears in training a reward proxy in an active learning setup, see [4], and we do think that it might be an interesting future extension of our work.

---

> ### Author Response · Authors · 2024-11-18
> **Author Response (Part 2)**
>
> [Question 4] Finally, combining constraint reinforcement learning with TLM is a valuable direction for further work since it allows us to explicitly incorporate challenging types of constraints, e.g., the correctness of the molecules in the QM9 task, in a more principled way. However, to the best of our knowledge, such approaches are yet to be explored in GFlowNet literature. That is why we leave this question as an interesting direction for further studies.
>
> References:
>
> [1] Sobhan Mohammadpour, Emmanuel Bengio, Emma Frejinger, Pierre-Luc Bacon. Maximum entropy GFlowNets with soft Q-learning. AISTATS 2024
>
> [2] Hyosoon Jang, Yunhui Jang, Minsu Kim, Jinkyoo Park, and Sungsoo Ahn. Pessimistic Backward Policy for GFlowNets. arXiv preprint arXiv:2405.16012
>
> [3] Emmanuel Bengio, Moksh Jain, Maksym Korablyov, Doina Precup, Yoshua Bengio. Flow Network based Generative Models for Non-Iterative Diverse Candidate Generation. NeurIPS 2021
>
> [4] Moksh Jain, Emmanuel Bengio, Alex-Hernandez Garcia, Jarrid Rector-Brooks, Bonaventure F. P. Dossou, Chanakya Ekbote, Jie Fu, Tianyu Zhang, Micheal Kilgour, Dinghuai Zhang, Lena Simine, Payel Das, Yoshua Bengio. Biological Sequence Design with GFlowNets, ICML 2022
>
> [5] Yihang Chen, Lukas Mauch. Order-Preserving GFlowNets. ICLR 2024

---

> > ### Comment · Reviewer_EZ4M · 2024-12-01
> >
> > (1) This is not an issue at the academic level. Although the paper validates the fundamental characteristics of TLM, it lacks a systematic and theoretically rigorous comparative analysis, particularly in discussing the strengths and weaknesses of other methods. This makes it difficult to fully comprehend the scientific contributions of TLM and its applicable scenarios. I believe this is an important point, and I agree with Reviewer 6Sem's assessment.
> >
> > (2) While the paper includes a discussion on the sEH task and the differences in TLM performance across various environments, it lacks a theoretical analysis to substantiate these observations.

---

> > > ### Author Response · Authors · 2024-12-01
> > >
> > > 1. We respectfully disagree with the reviewer’s evaluation. We remain committed to conducting a fair comparison with the paper [Jang et al., 2024]. However, the absence of source code and technical details for implementing the underlying method in the initial version of the paper available at the moment of ICLR 2025 deadline, which in turn requires detailed tuning, made it very challenging to perform a fair and comprehensive comparison. As we have already promised, and given that the authors of [Jang et al., 2024] have now released their source code, we will incorporate a comparison with this important baseline. Unfortunately, the limited discussion period clearly does not allow us to provide this comparison during rebuttal.
> > >
> > > 2. Furthermore, we would like to stress that our paper already includes a specific discussion of the methods proposed by the suggested closest competitors, [Jang et al., 2024] and [Mohammadpour et al., 2024]; see Section 2.3. In addition, we stress that in all our experiments, we provide the approach of [Mohammadpour et al, 2024] as one of the baselines, as well as argue that our results support some of the findings of [Mohammadpour et al, 2024], while empirically demonstrating the advantages of our method. If the reviewer believes that further discussion is required, we think that it is important that they could provide more specific aspects for their critics on the comparison to other literature. In its current form, the comment (1) appears too general and does not provide actionable guidance.
> > >
> > > 3. While in principle we could agree with the comment (2) and acknowledge that the TLM algorithm lacks theoretical explanation of its success in particular environments, this limitation applies to all known methodological papers addressing GFlowNet training. This is an open and challenging research problem, and if the reviewer could specify the particular type of theoretical analysis or guarantees they would like to see, we would be grateful and eager to address it. Otherwise, we also believe that this comment is not constructive enough to provide actionable guidance that we could incorporate to improve our paper.

---

> ### Comment · Area_Chair_o4MX · 2024-11-25
> **Please read rebuttal**
>
> Dear Reviewer EZ4M, Could you please read the authors' rebuttal and give them feedback at your earliest convenience? Thanks. AC

---

### Official Review · Reviewer_A6Yj · 2024-10-29

**Soundness:** 3
**Presentation:** 4
**Contribution:** 3
**Rating:** 8
**Confidence:** 4

**Summary:**

This paper studies the connection between GFlowNet training and entropy-regularized reinforcement learning problems–also known as soft RL. It specifically tackles this relationship in the setting of the fixed backward policy which is considered a significant limitation in the literature. For this indeed, they propose a simple backward policy optimization algorithm; Trajectory Likelihood Maximization (TLM)  that primarily involves a direct maximization of the value function in an entropy-regularized Markov Decision Process (MDP) over intermediate rewards. They formulate the GFlowNet training problem as a unified objective involving both forward and backward policies and propose an alternating minimization procedure. Their main contribution is deriving the first unified approach for adaptive backward policy optimization in soft RL-based GFlowNet methods, i.e. the TLM. They carry out an experimental evaluation on Hypergrid and Bit Sequence as well as two molecule design environments: sEH and QM9 to conclude that their algorithm converges faster and that can be integrated with any existing GFlowNet training algorithm. Lastly, there are empirical experiments done to show the performance of TLM under different hyperparameter choices.
In general, this paper could be a good contribution, with the caveat of some clarifications on the theory and experiments. Given these clarifications in an author's response, I would be willing to increase the score.

**Strengths:**

The authors study a very important problem in the GFlowNets and entropy-regularized RL literature which is backward policy optimization. They propose the TLM method that represents the first unified approach for adaptive backward policy optimization in soft RL-based GFlowNet methods which enhances mode exploration and accelerates convergence in complex GFlowNet environments. They formulate the GFlowNet training problem as a unified objective involving both forward and backward policies, proposing an alternating minimization procedure. For novelty, they approximate these two steps through a single stochastic gradient update and derive an adaptive approach for combining backward policy optimization with any GFlowNet method, including soft RL methods. The significance of their work arises from working on the limitations of two main previous research; Mohammadpour et al. (2024) and Jang et al. (2024). Furthermore, their approach can be connected to cooperative game theory by interpreting the forward policy PF as a forward player and the backward policy PB as a backward player, with both players attempting to minimize KL-divergence between corresponding trajectory distributions. For theory, they introduce a new theorem that shows that stability is essential for theoretical convergence. The quality of the proposed method is shown by their numerical experiments. TLM greatly speeds up mode discovery on the QM9 environment for all GFlowNet algorithms. Their results support the findings of Mohammadpour et al. (2024), where it was also observed that the backward policy optimization enhances GFlowNets on more complex and less structured environments like QM9. As a last note, the math behind their algorithm is neat and well-explained.

The main strengths are as follows:
- Deriving the trajectory likelihood maximization (TLM) method for backward policy optimization.
- The proposed method represents the first unified approach for adaptive backward policy optimization in soft RL-based GFlowNet methods. The method can be integrated with any existing GFlowNet training algorithm.
- Providing an extensive experimental evaluation of TLM in four tasks, confirming the findings of
Mohammadpour et al. (2024), emphasized the benefits of training the backward policy in complex environments with less structure.
- Providing the basis for implementing and extending their approach in different domains such as cooperative game theory.

**Weaknesses:**

The paper does not do a great job of demonstrating the deficiencies of the proposed algorithm. No section in this paper clearly explains and shows the difference between the proposed algorithm and other related work. In other words, the reader may not understand what specifically the work you are trying to build your work on. The authors point to references to establish closely related work such as Mohammadpour et al. (2024) and Jang et al. (2024), but do not clearly and plainly show what exactly they are improving, and what previous limitations they are trying to overcome. Furthermore, the authors mention that there are some similarities between their algorithm and a celebrated EM-algorithm (Dempster et al., 1977) and Hinton’s Wake-Sleep algorithm (Hinton et al., 1995), but did not mention what aspects those similarities are.
The experiments do not seem well justified, in the Hypergrid environment, Figure 1, the authors do not clearly explain why their method TLM fails to outperform the naive method in the TB training. In the Bit Sequences environment, the authors again do not show a sufficient explanation for the different results shown in Figure 2-Top row such as the improvement in DB and why TLM does not affect the results much in TB and SubTB. In addition, there seems to be no clear reason for producing results using two different metrics: Found Modes, and the Spearman Correlation between R and P_theta in three different environments, Bit sequences, sEH, and QM9. No explanation for Figure 2 (center and bottom) where TLM greatly speeds up mode discovery on QM9 for all GFlowNet algorithms, but shows similar or worse performance when compared to other backward approaches on sEH. There is no justified explanation of the poor performance of TLM on sEH in terms of mode discovery in all four Gflownets training algorithms.
Lastly, in Figure 3, their conclusion “TLM results in better correlations when paired with MunchausenDQN and SubTB, and shows similar results to the baselines when paired with DB and TB.” is not well justified. In what aspects does that affect your method? It would be much better to explain their method’s performance alongside its deficiencies in a more active discussion section.


Suggestions:
- Specify related work, similarities, and differences in a separate section.
- Explain your empirical results, i.e. your algorithm’s deficiencies and mention suggestions to enhance its performance.

Minor comments:
1. Appendix. A.1 Omitted Proofs P. 14 Line 2, “by the choice of rewards”

**Questions:**

For the related work, I still cannot understand what connections you are trying to make and which research you are trying to use and build upon. More specifically, what limitations are you trying to overcome? What are the similarities between your algorithm and the celebrated EM-algorithm (Dempster et al., 1977) and Hinton’s Wake-Sleep algorithm (Hinton et al., 1995) and how does TLM differ from their algorithm?
The experimental work does not seem well justified, but there is no discussion of the varying performances across the environments; what conclusions are you deriving? Do you have any explanations for the deficiencies your algorithm has which were shown in some parts of the empirical results? Do you have any suggestions in order to overcome those shortages?
The experiments do not provide convincing evidence of the correctness of the proposed approach or its utility compared to existing approaches. There are so many missing details it is difficult to draw many conclusions:
- Why did you use an online backward policy to compute the loss for the backward policy L_TLM?
- Why did you use a target backward policy to compute the loss for the forward policy L_Alg?
- In Hypergrid, all models are parameterized by MLP with 2 hidden layers, why? There seems no study of the decaying learning rate conducted on the Hypergrid environment, any comments?
- Why did you choose transformers to parameterize your models in the Bit Sequences environment?
- There is no clear rule for choosing your hyperparameters; it appears that you are simply taking the values established in previous related work. Why?
The differences in the “full plots” compared to the original ones are very small when choosing decaying learning rates. Can you provide some context to understand the significance of this difference?

---

> ### Author Response · Authors · 2024-11-18
> **Author Response (Part 1)**
>
> We want to express our gratitude to the reviewer for the very detailed review, the time and effort spent reading our paper, the positive feedback, and the valuable suggestions.
>
> First, we would like to stress one of the main points of our work outlined in our Introduction: considering backward policy optimization in GFlowNets from the RL perspective and closing the gap related to it in the theory that connects GFlowNets and RL. The existing theory previously only considered the setting of a fixed backward policy. This point was never explored in the previous works, so the novelty and the motivation for our TLM algorithm directly arise from it. [1] utilizes RL as a tool to find their maxent backward policy, and the connection to RL is made only in the case of using this exact unique maxent policy. [7] does not consider the RL perspective at all. The scope of our work is to explain the simultaneous optimization of the forward and backward policies from the RL perspective and empirically show the benefits of our TLM algorithm, which is grounded in this perspective. We thank the reviewer for pointing out this weakness in the presentation. We added a corresponding clarification to Section 2.3.
>
> We also highly appreciate the comments about unclear discussions and a lack of explanations of some experimental results. As suggested, we added a more detailed Discussion subsection to the end of Section 4 to improve the clarity regarding these points, where we consider two hypotheses. In regard to the poor performance on sEH, we argue that our experimental results are in line with the hypothesis put out by [1], which states that backward policy optimization is more beneficial in environments with less structure, like QM9. While for highly structured tasks like sEH, considering non-trivial backward approaches is of little utility and using a simple uniform backward policy generally shows strong performance. Regarding the performance differences when TLM is combined with various forward policy training objectives, we hypothesize that local GFlowNet training objectives defined on individual transitions, e.g., DB, benefit more from the combination with TLM. TLM propagates global information over whole trajectories, resulting in a good synergy with such local objectives. Meanwhile, TB, which is also defined on whole trajectories, shows fewer improvements when combined with TLM. We refer the reviewer to the updated manuscript for a more detailed discussion.
>
> As for the metrics, reward correlation and the number of found modes are standard metrics that are widely utilized in GFlowNet literature, especially in molecule generation environments [1,3,4,5]. The use of two metrics is justified by the different aspects they are designed to measure:
> 1. Reward correlation measures how well the distribution over the target space induced by the trained forward policy matches the target reward distribution, i.e., how well the sampling problem is solved.
> 2. The number of found modes measures how many diverse high-reward objects the model discovers during training, which is the quantity we are interested in scientific applications but the one that does not directly measure how well we fit the target distribution, rather focusing on the exploration of the environment.
>
> Regarding the similarities between our approach, the EM algorithm, and the Wake-Sleep Algorithm, we meant the structure of alternating minimization. This remark was made to underline that such a way to address the problem with a natural block structure is well-known in the literature. We would be happy to incorporate any changes the reviewer might suggest or even remove this remark.
>
> As for the target networks for backward policies, it is a standard technique from reinforcement learning literature [2] utilized to introduce stability into the training procedure (which is required by our Theorem 3.1 and additionally explained in Appendix A.2). Online network is exactly the backward policy network that is trained via TLM objective. The simplest intuition behind using the target network in algorithms similar to Q-learning is that it prevents the target values from changing too fast, stabilizing the training process. Toward this aim, we maintain a separate, slowly updated copy of the backward policy network. Some theoretical results (see, e.g. [6]) provide theoretical support for the fact that “a target network stabilizes training” using a framework of the two-timescale stochastic approximation. However, we acknowledge that this fact is a common empirical wisdom rather than a theoretically established result.

---

> ### Author Response · Authors · 2024-11-18
> **Author Response (Part 2)**
>
> As for the hyperparameter and neural network architecture choices, for the most part, we indeed utilize the choices established in previous literature [3, 4, 5] to follow their setups faithfully. Note that the goal of our experimental evaluation is not necessarily to obtain the best possible metrics in all of the tasks (which could be potentially improved in many ways, e.g., by taking larger neural networks) but rather to produce a direct comparison of various approaches to selecting the backward policy in a fair and controlled setup.
>
> We thank the reviewer for the questions about plots with varying learning rates and their connection to the full plots in Appendix, which indeed needed to be discussed in the paper in sufficient detail. For bit sequence and molecule generation tasks, we consider the initial learning rate as a hyperparameter and study the dependence of the results for all methods on the choice. Figure 3 does not present the results for single training runs depending on the learning rate decay across the run, but the final results for different training runs with varying initial learning rates (depicted on the horizontal axis). Figure 2, on the other hand, illustrates how the number of discovered modes increases over the training duration for runs with fixed initial learning rates, while curves depending on the varying initial learning rates are presented in full plots in the Appendix in Figure 5 and Figure 6. This allows one to directly examine the robustness of various approaches with respect to the choice of learning rate, thus having a benefit over the type of presentation where one simply chooses the best learning rate for each method.  We note that such a presentation was also used in some of the previous works [4].
>
> References:
>
> [1] Sobhan Mohammadpour, Emmanuel Bengio, Emma Frejinger, Pierre-Luc Bacon. Maximum entropy GFlowNets with soft Q-learning. AISTATS 2024
>
> [2] Volodymyr Mnih, Koray Kavukcuoglu, David Silver, Andrei A Rusu, Joel Veness, Marc G Bellemare, Alex Graves, Martin Riedmiller, Andreas K Fidjeland, Georg Ostrovski, et al. Human-level control through deep reinforcement learning. Nature, 518, 2015
>
> [3] Nikolay Malkin, Moksh Jain, Emmanuel Bengio, Chen Sun, Yoshua Bengio. Trajectory balance: Improved credit assignment in GFlowNets. NeurIPS 2022
>
> [4] Kanika Madan, Jarrid Rector-Brooks, Maksym Korablyov, Emmanuel Bengio, Moksh Jain, Andrei Nica, Tom Bosc, Yoshua Bengio, Nikolay Malkin. Learning GFlowNets from partial episodes for improved convergence and stability. ICML 2023
>
> [5] Daniil Tiapkin, Nikita Morozov, Alexey Naumov, Dmitry Vetrov. Generative Flow Networks as Entropy-Regularized RL. AISTATS 2024
>
> [6] Shangtong Zhang, Hengshuai Yao, Shimon Whiteson. Breaking the Deadly Triad with a Target Network. ICML 2021
>
> [7] Hyosoon Jang, Yunhui Jang, Minsu Kim, Jinkyoo Park, and Sungsoo Ahn. Pessimistic Backward Policy for GFlowNets. arXiv preprint arXiv:2405.16012

---

> > ### Comment · Reviewer_A6Yj · 2024-11-26
> >
> > Thank you for your detailed responses to my review. After reading your responses, your revisions, and other reviews, I decided to maintain my score. I tend to accept this paper since it tackles an important aspect and provides an interesting method.

---

> ### Comment · Area_Chair_o4MX · 2024-11-25
> **Please read rebuttal**
>
> Dear Reviewer A6Yj, Could you please read the authors' rebuttal and give them feedback at your earliest convenience? Thanks. AC

---

### Official Review · Reviewer_6Sem · 2024-11-01

**Soundness:** 3
**Presentation:** 3
**Contribution:** 3
**Rating:** 5
**Confidence:** 4

**Summary:**

This paper introduces a method called Trajectory Likelihood Maximization (TLM) for optimizing backward policies in Generative Flow Networks (GFlowNets). The authors formulate the GFlowNet training problem as a unified objective involving both forward and backward policies and propose an alternating minimization procedure. The method is validated through extensive experimental evaluation across various benchmarks.

**Strengths:**

- The proposed TLM method is designed to be easily implementable and compatible with existing GFlowNet training algorithms.
- The authors validate their approach across four different tasks, demonstrating its effectiveness in various scenarios.
- The paper is well-structured and clearly written, with careful explanations of technical concepts and methodological choices.

**Weaknesses:**

- Limited comparative analysis: While the paper acknowledges recent work by Jang et al. (2024) on pessimistic backward policies, it doesn't provide a direct experimental comparison with this highly relevant approach. Such a comparison would help readers understand the relative advantages of TLM. Though the experiments are extensive, it would be valuable to see more detailed analysis of how TLM performs compared to other backward policy optimization methods across different types of environments.
- Ablation studies: The paper could benefit from more detailed ablation studies to understand the contribution of different components of the proposed method.
- Results presentation: It will be better to combine the last three columns in Figure 1 to a single figure as they are just different variants of GFlowNets training objectives. Including the best-performing loss will better present the results (e.g., row 2).

Hyosoon Jang, Yunhui Jang, Minsu Kim, Jinkyoo Park, and Sungsoo Ahn. Pessimistic backward policy for gflownets. arXiv preprint arXiv:2405.16012, 2024.

**Questions:**

1. Could the authors provide a direct experimental comparison with the pessimistic backward policy approach proposed by Jang et al. (2024)? This would help establish the relative merits of TLM in similar settings.
2. Can the authors provide an explanation for the performance of TLM with different choices of the GFlowNet training objective (TB, DB, SubTB)? Are there certain objectives where TLM shows particularly strong or weak performance?

---

> ### Author Response · Authors · 2024-11-18
>
> We thank the reviewer for their valuable suggestions and are happy to provide further details in response to their concerns.
>
> We agree with the importance of comparative evaluation to previous works. However, we note that at the moment of the ICLR 2025 submission deadline, the pessimistic backward policy paper [1] had not yet made its code publicly available, and the implementation details provided in the arXiv preprint were limited, making it challenging to reproduce the presented approach. After the ICLR 2025 submission deadline, the authors published the code for their method and experiments. Although it will be unfeasible to make a careful and faithful comparison during the time allocated for rebuttal, we are more than willing to include a comprehensive comparison to the approach of [1] in the camera-ready version of our paper.
>
> Second, we note that we do present an ablation study of different components of our method in Appendix A.2. We try to separately turn off various components of TLM, e.g. backward policy target networks, and experimentally verify that they have a positive effect on the performance.
>
> Then, we would like to disagree about the presentation in Figure 1. DB, TB, and SubTB are indeed different existing GFlowNet objectives we utilize to train the forward policy, each having its own merits and performing better or worse between different tasks. One of the points of our experimental evaluation was to investigate how backward policy training with TLM would perform when paired with different forward policy training objectives, which can provide valuable insights for understanding our method and its utility. Just showing the best-performing forward policy training method would conceal some relevant information from the reader, and we note this as a limitation of the experimental evaluation provided in previous works on backward policy training [1, 2].
>
> Finally, we highly appreciate the suggestion to provide a more detailed explanation for the performance of TLM with different choices of the GFlowNet training objective, which is related to our previous point presented above. We added a detailed discussion of our experimental results to the end of Section 4 of our paper. In summary, we hypothesize that local GFlowNet training objectives defined on individual transitions, e.g., DB, benefit more from the combination with TLM, which in turn propagates global information over whole trajectories, resulting in a good synergy. Meanwhile, TB, which is also defined on whole trajectories, shows fewer improvements when combined with TLM. We refer the reviewer to the updated manuscript for a more detailed discussion.
>
> References:
>
> [1] Hyosoon Jang, Yunhui Jang, Minsu Kim, Jinkyoo Park, and Sungsoo Ahn. Pessimistic Backward Policy for GFlowNets. arXiv preprint arXiv:2405.16012
>
> [2] Sobhan Mohammadpour, Emmanuel Bengio, Emma Frejinger, Pierre-Luc Bacon. Maximum entropy GFlowNets with soft Q-learning. AISTATS 2024

---

> ### Comment · Area_Chair_o4MX · 2024-11-25
> **Please read rebuttal**
>
> Dear Reviewer 6Sem, Could you please read the authors' rebuttal and give them feedback at your earliest convenience? Thanks. AC

---

### Official Review · Reviewer_q9aZ · 2024-11-04

**Soundness:** 3
**Presentation:** 3
**Contribution:** 2
**Rating:** 6
**Confidence:** 3

**Summary:**

This paper proposes a novel method to optimize the backward policy of the GFlowNet during training. The proposed method can be combined with any GFlowNet training method, and the experimental results demonstrate that it can bring performance improvement in some tasks.

**Strengths:**

- This paper is well written. The presentation and structure of this paper are clear.
- The proposed method, which employs an EM-style mechanism to optimize the training objective, is interesting. Theoretical analysis guarantees its convergence, and it can be used in any GFlowNet algorithm.

**Weaknesses:**

- The performance improvement is marginal. On the sEH task, the proposed method even harms the performance, as explained in line 485. Moreover, the experiments (e.g., Qm9 and sEH) included in this paper are somewhat toy. I encourage the authors to consider more complex and large-scale benchmarks, such as AMP and RMA generation [1].


[1] Biological Sequence Design with GFlowNets, ICML 2022

**Questions:**

NA

---

> ### Author Response · Authors · 2024-11-18
>
> We thank the reviewer for their positive feedback and are happy to provide further details in response to their concerns.
>
> First, we would like to disagree with the comment about the toy nature of sEH and QM9 tasks. sEH is a challenging molecule generation environment, introduced in the seminal work on GFlowNets [1] and used as a benchmark in many GFlowNet papers, with the number of states $\approx 10^{16}$ and number of actions from $100$ to $2000$ (see [1], Section 4.2). There exists a simplified version of the environment in GFlowNet literature with a reduced number of fragments and thus reduced state space size, see, e.g. [2, 6]. However, we emphasize that this is not the setting considered in our submission. As for the QM9 task, it can be small and toy if the generation is done in a fragment-based fashion, as in [2]. However, we do the generation atom-by-atom and bond-by-bond, as in [3], which results in a more challenging task with a larger and less structured state space. We can approximate the number of states in it to be $\approx 4.5 \cdot 10^{14}$ and the number of actions to vary from $6$ to $100$.
>
> In regard to the poor performance on sEH, we argue that our experimental results are in line with the hypothesis put out by [3], which states that backward policy optimization is more beneficial in environments with less structure, like atom-based QM9. While for highly structured tasks like sEH, considering non-trivial backward approaches is of little utility and using a simple uniform backward policy generally shows strong performance. We added a more detailed Discussion subsection to the end of Section 4 in our paper to improve the clarity regarding this point.
>
> Second, we appreciate the suggestion to extend our results to other benchmarks. However, we would like to note that the suggested biological sequence generation environments from [4] are autoregressive, meaning that all possible actions are to append a token to the end of the sequence. This results in a tree-structured environment where each state has exactly one parent; thus, only one trivial choice of a backward policy exists, and its training is redundant. Other works on GFlowNets have made attempts at considering non-tree structured environments for biological sequence generation [5], but such environments are not well-established yet in the GFlowNet literature. We would like to highlight that the careful design of a new environment is a challenging problem, which is of independent interest and requires a lot of effort. That is why we consider it to be an interesting further research direction.
>
> References:
>
> [1] Emmanuel Bengio, Moksh Jain, Maksym Korablyov, Doina Precup, Yoshua Bengio. Flow Network based Generative Models for Non-Iterative Diverse Candidate Generation. NeurIPS 2021
>
> [2] Max W. Shen, Emmanuel Bengio, Ehsan Hajiramezanali, Andreas Loukas, Kyunghyun Cho, Tommaso Biancalani. Towards Understanding and Improving GFlowNet Training. ICML 2023
>
> [3] Sobhan Mohammadpour, Emmanuel Bengio, Emma Frejinger, Pierre-Luc Bacon. Maximum entropy GFlowNets with soft Q-learning. AISTATS 2024
>
> [4] Moksh Jain, Emmanuel Bengio, Alex-Hernandez Garcia, Jarrid Rector-Brooks, Bonaventure F. P. Dossou, Chanakya Ekbote, Jie Fu, Tianyu Zhang, Micheal Kilgour, Dinghuai Zhang, Lena Simine, Payel Das, Yoshua Bengio. Biological Sequence Design with GFlowNets, ICML 2022
>
> [5] Kanika Madan, Jarrid Rector-Brooks, Maksym Korablyov, Emmanuel Bengio, Moksh Jain, Andrei Nica, Tom Bosc, Yoshua Bengio, Nikolay Malkin. Learning GFlowNets from partial episodes for improved convergence and stability. ICML 2023
>
> [6] Yihang Chen, Lukas Mauch. Order-Preserving GFlowNets. ICLR 2024

---

> > ### Comment · Reviewer_q9aZ · 2024-11-22
> >
> > Thanks for your responses to my review. After reading your responses and other reviews, I decided to maintain my score. I tend to accept this paper since it provides an interesting method that could encourage future research. However, as pointed out by other reviewers, the authors should incorporate the revisions and additional experiments (compared with the previous backward-policy optimization method ) in their final version.

---

### Author Response · Authors · 2024-11-18
**Paper Revision**

We highly appreciate the valuable feedback provided by all reviewers. We are pleased that reviewers acknowledged the novelty of our TLM approach for backward policy optimization, the importance of its theoretical grounds, and the fact that it is compatible with various existing GFlowNet forward policy training algorithms. Moreover, reviewers q9aZ, 6Sem, EZ4M suggested that the presentation of the main ideas of the paper is clear. At the same time, we agree with suggestions to revise the discussion and analysis of our experimental results and appreciate the feedback on it. Within the allowed time, we made a revision to the paper following the feedback from reviewers, which we believe further improves our submission.
1. We added a detailed Discussion subsection to the end of Section 4 of our manuscript, where we carefully examine the empirical results and discuss two hypotheses that can provide important insight. The first explains the difference in performance of TLM between sEH and QM9 molecule generation environments, and the second explains the differences in TLM performance when it is combined with different forward policy training algorithms.
2. In addition, we added some small remarks to Section 2.3 and Section 4 based on the feedback from reviewer A6Yj.

All changes in the new pdf are highlighted in blue.

---

### Meta-Review · Area_Chair_o4MX · 2024-12-19

**Metareview:**

This paper proposes to optimize the backward policy of the GFlowNet. The proposed method is the GFlowNet training method, and the experimental results demonstrate that it can improve performance. This approach has a reasonable theoretical foundation and shows a unified framework for policy optimization. The paper is clearly written and well presented. The paper is above the bar of ICLR.

**Additional Comments On Reviewer Discussion:**

Most of the reviewers' concerns are resolved.

---

### Decision · Program_Chairs · 2025-01-22

Accept (Poster)